# Genes encoding cytochrome P450 monooxygenases and glutathione S-transferases associated with herbicide resistance evolved before the origin of land plants

**Alexandra Casey[1,2], Liam Dolan [1,2]***

**1** Gregor Mendel Institute, Vienna, Austria, **2** Department of Plant Sciences, University of Oxford, Oxford, Oxfordshire, United Kingdom

* liam.dolan@gmi.oeaw.ac.at

**Data Availability Statement:** All relevant data are within the manuscript and its Supporting Information files.

## Abstract

Cytochrome P450 (CYP) monooxygenases and glutathione S-transferases (GST) are enzymes that catalyse chemical modifications of a range of organic compounds. Herbicide resistance has been associated with higher levels of CYP and GST gene expression in some herbicide-resistant weed populations compared to sensitive populations of the same species. By comparing the protein sequences of 9 representative species of the Archaeplastida–the lineage which includes red algae, glaucophyte algae, chlorophyte algae, and streptophytes–and generating phylogenetic trees, we identified the CYP and GST proteins that existed in the common ancestor of the Archaeplastida. All CYP clans and all but one land plant GST classes present in land plants evolved before the divergence of streptophyte algae and land plants from their last common ancestor. We also demonstrate that there are more genes encoding CYP and GST proteins in land plants than in algae. The larger numbers of genes among land plants largely results from gene duplications in CYP clans 71, 72, and 85 and in the GST phi and tau classes [1,2]. Enzymes that either metabolise herbicides or confer herbicide resistance belong to CYP clans 71 and 72 and the GST phi and tau classes. Most CYP proteins that have been shown to confer herbicide resistance are members of the CYP81 family from clan 71. These results demonstrate that the clan and class diversity in extant plant CYP and GST proteins had evolved before the divergence of land plants and streptophyte algae from a last common ancestor estimated to be between 515 and 474 million years ago. Then, early in embryophyte evolution during the Palaeozoic, gene duplication in four of the twelve CYP clans, and in two of the fourteen GST classes, led to the large numbers of CYP and GST proteins found in extant land plants. It is among the genes of CYP clans 71 and 72 and GST classes phi and tau that alleles conferring herbicide resistance evolved in the last fifty years.

**Funding:** This research was supported by a European Research Council (ERC) Advanced Grants EVO500 project number 250284 and De Novo-P (project number 787613) to LD from the European Commission. AC was supported by a British Biological Sciences Research Council (BBSRC) Scholarship through a doctoral training partnership (BB/M011224/1). The funders had no role in study design, data collection and analysis, decision to publish, or preparation of the manuscript.

**Competing interests:** L. D. is a co-founder of moa Technology Ltd. This does not alter our adherence to PLOS ONE policies on sharing data and materials.

## Introduction

Herbicide resistance evolves in weed populations and poses a challenge in all agricultural landscapes where chemical herbicides are used for weed control. This resistance can result from two types of mutations. Mutations in the gene targeted by the herbicide which reduce the affinity of the herbicide for the target site confer target site resistance (TSR). Non target site resistance (NTSR) results either from mutations that reduce the amount of herbicide chemical reaching the target or that alleviate herbicide-induced damage [3]. Reported mechanisms of NTSR involve reduced herbicide uptake or translocation, herbicide metabolism or sequestration [4–6]. Genetic changes in genes encoding enzymes that can metabolise the herbicide, such as overexpression resulting from gene amplification or mutations in regulatory regions, can inactivate the herbicide, conferring resistance [7,8]. While the genetic basis of NTSR is often complex and mechanistically poorly understood, the overexpression of genes encoding cytochrome P450 monooxygenases and glutathione *S*-transferases has been shown to confer resistance in weed populations [9–11].

Glutathione-*S*-transferases (GSTs) are an ancient superfamily of enzymes found in eukaryotes and prokaryotes. GSTs catalyse the conjugation of glutathione (GSH) to both endogenous and exogenous electrophilic, hydrophobic substrates to form more polar, hydrophilic compounds. GSTs also catalyse GSH-dependent peroxidase, isomerase, and deglutathionylation reactions. In plants, GSTs are active in diverse processes including abiotic and biotic detoxification pathways [12,13], ascorbic acid metabolism [14], hormone signalling such as auxin and cytokinin homeostasis [15–17], metabolism of anthocyanins and flavonoids [18,19], tyrosine catabolism [20], and in preventing apoptosis [21].

GSTs function as either monomers or dimers. Each monomer is characterised by a conserved N-terminal domain containing the active site and several GSH binding site residues (G-sites), and a less conserved C-terminal domain comprising alpha helices with class-specific substrate binding sites (H-sites) [22–24]. Plant GSTs are classified into groups as cytosolic, mitochondrial, or microsomal and each group is further subdivided into classes based on sequence identity and kinetic properties [25]. In plants there are 12 cytosolic GST classes. These include tau (GSTU) [22,26], phi (GSTF) [22], theta (GSTT) [22,26], lambda (GSTL) [14], zeta (GSTZ) [22,27], iota (GSTI) [28], hemerythrin (GSTH) [28], tetrachlorohydroquinone dehalogenase (TCHQD) [28], eukaryotic translation elongation factor 1B-γ subunit (Ef1Bγ) [29], ureidosuccinate transport 2 prion protein (Ure2p) [28], glutathionyl hydroquinone reductase (GHR) [30], and dehydroascorbate reductase (DHAR) [14]. There is one microsomal GST class, microsomal prostaglandin E-synthase type 2 (mPGES2) [31,32], and one mitochondrial GST class, metaxin (GSTM) [33].

Cytochrome p450 monooxygenases (CYPs) are a superfamily of membrane-bound enzymes present in plants, fungi, bacteria, and animals. They are heme-thiolate proteins that use molecular oxygen and NADPH to modify substrates with diverse chemical reactions including oxidations, hydroxylations, dealkylations, and reductions [34] and are implicated in a wide array of biochemical pathways. CYPs participate in the synthesis and modification of primary metabolites such as sterols and fatty acids, secondary metabolites such as phenylpropanoids, glucosinolates, and carotenoids, and the synthesis and catabolism of hormones such as gibberellins, jasmonic acid, abscisic acid, brassinosteroids, and strigolactones [34–36].

CYPs are characterised by a conserved heme-binding domain, an oxygen binding domain, two conserved motifs (X-E-X-X-R and P-E-R-F) that form what is known as the ERR triad and is involved in positioning and stabilising the heme pocket, and several highly variable substrate positioning and recognition sites [37]. The three-dimensional structure of CYPs is conserved across the family even though the amino acid sequences of individual members may be

as little as 20% identical [37–39]. Previous phylogenetic analyses of CYPs grouped them into monophyletic clades termed clans, each containing one or more CYP families [40–42]. Sequences with more than 40% amino acid sequence identity are grouped within the same family, and those with more than 55% identity are grouped in the same subfamily [43]. Families are designated by numbers, and subfamilies by a letter after the number. Clans are named after their lowest numbered family member [43,44]. Clans represent the deepest clades that reproducibly appear in multiple phylogenetic trees.

Here we report the phylogenetic relationships among both the GST and CYP proteins within the Archaeplastida lineage. The Archaeplastida are an ancient monophyletic group of eukaryotes that contain plastids derived from a primary endosymbiotic event [45–47]. They include glaucophyte algae (algae that lack chlorophyll b and contain plastids surrounded by a vestigial peptidoglycan layer), rhodophyte algae (red algae that lack chlorophyll b and rely on accessory pigments for photosynthesis), chlorophyte algae (green algae that contain chlorophyll a and b but lack certain proteins found in streptophytes) and streptophytes (streptophyte algae and land plants) [48–51]. We showed that those CYPs and GSTs that have been shown to confer herbicide resistance among weeds are restricted to two monophyletic clans and two monophyletic classes, respectively. These clans and classes already existed in the common ancestor of land plants (embryophytes), which is estimated to have existed between 980 and 473 million years ago (Mya) [52–54]. These clans and classes diversified early in embryophyte evolution and now constitute the largest groups of CYP and GST proteins in extant vascular plants. This analysis suggests that natural selection caused by herbicides acts on sets of ancient genes that existed in the last common ancestor of the land plants and *Klebsormidium nitens*, a streptophyte alga, and diversified in vascular plants, leading to the evolution of herbicide resistance in the agricultural landscape.

## Materials and methods

### Data resources

Protein sequences from *Arabidopsis thaliana* were retrieved from TAIR10 [55] (https://www.arabidopsis.org/). Protein sequences from *Oryza sativa* ssp. *japonica* were retrieved from the rice genome annotation project [56] (http://rice.plantbiology.msu.edu/).Protein sequences from the liverwort *Marchantia polymorpha* were obtained from MarpolBase (http://marchantia.info/). Protein sequences from the hornwort *Anthoceros agrestis* were obtained from [57] (https://www.hornworts.uzh.ch/en/download.html). Protein sequences from the streptophyte alga *Klebsormidium nitens* were obtained from the *K. nitens* genome webpage [58] (http://www.plantmorphogenesis.bio.titech.ac.jp/~algae_genome_project/klebsormidium/). Protein sequences from the chlorophyte alga *Chlamydomonas reinhardtii*, the moss *Physcomitrium patens* and the lycophyte *Selaginella moellendorffii* were retrieved from Phytozome 12 [59] (https://phytozome.jgi.doe.gov/pz/portal.html). Protein sequences from the red alga *Cyanidioschyzon merolae* were retrieved from the *C. merolae* genome webpage [60] (https://www.genome.jp/kegg-bin/show_organism?org=cme).

A classification of CYP genes from *A. thaliana*, *S. moellendorffii*, *P. patens*, *C. reinhardtii* is available on The Cytochrome P450 Homepage [44] (http://drnelson.uthsc.edu/plants/). Two other Arabidopsis CYP databases can be found on the Arabidopsis Cytochrome P450 List [61] (http://www.p450.kvl.dk/At_cyps/table.shtml) and CyPEDIA [62] (http://www.ibmp.u-strasbg.fr/~CYPedia/). The classification of *O. sativa* CYPs is available on the University of California, Davis Rice CYP Database (https://ricephylogenomics.ucdavis.edu/p450/).

## Sequence collection

CYP protein sequences from *A. thaliana* and *O. sativa* [63,64] were used to perform BLASTP searches using a minimum E value cut-off of $1e^{-10}$ against the predicted proteomes of *S. moellendorffii*, *M. polymorpha*, *A. agrestis*, *P. patens*, *K. nitens*, *C. reinhardtii*, and *C. merolae*. GST protein sequences were retrieved by BLASTP searches using GST proteins from *A. thaliana* [65,66], *O. sativa* [67,68], and *P. patens* [28] against the predicted proteomes of *S. moellendorffii*, *M. polymorpha*, *A. agrestis*, *K. nitens*, *C. reinhardtii*, and *C. merolae*. This initial list of sequences for each species was used as a query for BLASTP searches against the proteome of that species to retrieve additional sequences belonging to species-specific clans. Each CYP sequence was checked for the presence of the cytochrome p450 domain (PF00067, IPR00128) and each GST sequence was checked for the presence of the GST N-terminal domain (IPR004045, IPR019564, PF13409, PF17172, PF13417 and PF02798) and C-terminal domain (IPR010987, PF13410, PF00043, PF14497 and PF17171) using InterProScan 84.0 [69].

Two enzyme families with glutathione transferase activity, kappa [70] and membrane associated proteins in eicosanoid and glutathione metabolism (MAPEG) [71], do not possess a GST N-terminal thioredoxin-like domain or GST C-terminal domain and lack the N-terminal active site found in all other GST proteins. An additional group of sequences was identified by this analysis possessing two GST N-terminal domains (2N) but lacking a C-terminal domain. Protein sequences belonging to the kappa, MAPEG, and 2N classes were therefore not included in the phylogenetic analysis but are listed in S5 Table.

## Sequence alignment

Sequences were aligned in MAFFT [72] using the FFT-NS-2 algorithm and visualised in Bioedit [73]. Sequences lacking important functional residues were removed. Important residues in plant CYP and GST sequences are indicated in S1, S3 and S4 Figs. To trim large gaps, four approaches to alignment cleaning were undertaken. A manual approach was carried out using knowledge of the location of the functionally important CYP and GST residues. A more stringent trimming approach was also tested with the trimming software trimAl v.1.2. [74] using the three automated modes (-gappyout, -strict and -strictplus) (S2 Fig). For both the GST and CYP phylogenetic trees, the approximate likelihood ratio test (aLRT) support values for the deepest clades of the maximum-likelihood (ML) trees resulting from the trimAl -strict and -strictplus alignments were low (0–0.23) (S2 Fig). The ML trees generated from the trimAI -gappyout alignments had correct tree topologies but had low aLRT support values for the main clades (0.05–0.23). The ML trees generated from the manually trimmed GST and CYP alignments had the overall highest aLRT values (>0.8) for the main clades and were selected as the representative trees for further analysis.

## Phylogenetic analysis

The final alignments were subjected to a maximum-likelihood analysis conducted by PhyML 3.0 [75] using an estimated gamma distribution parameter, the LG+G+F model of amino acid substitution, and a $Chi^2$-based approximate likelihood ratio test (aLRT). The resulting unrooted trees were visualised in Figtree v1.4.4 [76] and annotated in Inkscape v1.0.2 [77].

## Results

1130 CYP and 358 GST sequences were identified in the genomes of 9 species of Archaeplastida

**Table 1. List of species used in the analysis.**

| Classification | | | | | Clade | Species | Genome (Mb) | Protein-coding genes | GSTs | GSTs (% PCG) | CYPs | CYPs (% PCG) | References |
|---|---|---|---|---|---|---|---|---|---|---|---|---|---|
| | | | | | Angiosperm eudicot | *Arabidopsis thaliana* | 135 | 25,498 | 61 | 0.24 | 238 | 0.93 | [78] |
| | | | | 8 | Angiosperm monocot | *Oryza sativa* | 321 | 35,681 | 85 | 0.24 | 291 | 0.82 | [56] |
| | | | | | Lycophyte | *Selaginella moellendorffii* | 212.6 | 22,285 | 57 | 0.26 | 199 | 0.89 | [79] |
| | 3 | 5 | 7 | | Hornwort | *Anthoceros agrestis* | 133 | 24,700 | 26 | 0.11 | 144 | 0.58 | [57] |
| 1 | | | | 9 | Moss | *Physcomitrium patens* | 480 | 35,938 | 42 | 0.12 | 69 | 0.19 | [80] |
| | | | | | Liverwort | *Marchantia polymorpha* | 225.8 | 19,138 | 35 | 0.18 | 115 | 0.60 | [81] |
| | | 6 | | | Streptophyte alga | *Klebsormidium nitens* | 117.1 | 16,215 | 24 | 0.15 | 29 | 0.18 | [58] |
| | 4 | | | | Chlorophyte alga | *Chlamydomonas reinhardtii* | 120 | 15,143 | 19 | 0.13 | 40 | 0.26 | [82] |
| | 2 | | | | Rhodophyte alga | *Cyanidioschyzon merolae* | 16.5 | 5,331 | 9 | 0.17 | 5 | 0.09 | [60] |

Including their position in the Archaeplastida (Classification) in which 1 = Archaeplastida, 2 = Rhodophyta (red algae), 3 = Viridiplantae (green plants), 4 = Chlorophyta, 5 = Streptophyta, 6 = Charophyta (streptophyte algae), 7 = Embryophyta (land plants), 8 = Tracheophyta (vascular plants), 9 = Bryophyta (non-vascular plants); their subgroup (Clade); genome size (Genome); total number of protein-coding genes (Protein-coding genes); total number of GST proteins (GSTs); GST proteins as a percentage of total protein coding genes (GSTs % PCG); total number of CYP proteins (CYPs); CYP proteins as a percentage of total protein coding genes (CYPs % PCG) and the bibliographical reference for each genome sequence.

To determine the phylogenetic relationships among CYP and GST sequences in the Archaeplastida lineage, we collected sequences from online resources. CYP and GST protein-coding genes in 9 species (Table 1) representing key Archaeplastida lineages were identified as described in Methods. The resulting 1130 CYP and 358 GST sequences included sequences from the red alga *Cyanidioschyzon merolae* (5 CYP and 9 GST proteins), the chlorophyte alga *Chlamydomonas reinhardtii* (40 CYP and 19 GST proteins), the streptophyte alga *Klebsormidium nitens* (29 CYP and 24 GST proteins), the liverwort *Marchantia polymorpha* (115 CYP and 35 GST proteins), the moss *Physcomitrium patens* (69 CYP and 42 GST proteins), the hornwort *Anthoceros agrestis* (144 CYP and 26 GST proteins), the lycophyte *Selaginella moellendorffii* (199 CYP and 57 GST proteins), and the angiosperms *Oryza sativa* (291 CYP and 85 GST proteins) and *Arabidopsis thaliana* (238 CYP and 61 GST proteins) (Table 1). On average, 126 CYP sequences and 40 GST sequences were identified in each species. The *M. polymorpha* CYP sequences were named following the standard CYP nomenclature [43]. These species were selected as representative species for lineages in which there was high quality full genome information. In addition, the availability of a CYP classification for *A. thaliana*, *O. sativa*, *S. moellendorffii*, *P. patens* and *C. reinhardtii* CYP genes facilitated the clan classification of CYPs identified in the other species in this study.

## Plant CYP clans are ancient and two CYP clans existed in the last common ancestor of the Archaeplastida

To elucidate the evolution of CYPs in Archaeplastida, we constructed a phylogenetic tree using a maximum likelihood approach (Fig 1A). This analysis demonstrated that CYPs from the 9 representative species of Archaeplastida grouped into 17 monophyletic clans, consistent with previous analyses of plant CYP phylogeny [40–42].

CYPs encoded by the genomes of land plants *A. agrestis*, *M. polymorpha*, *P. patens*, S. *moellendorffii*, *O. sativa*, or *A. thaliana* corresponded to 12 of the 17 clans identified in the Archaeplastida– 51, 71, 72, 74, 85, 86, 97, 710, 711, 727, 746, and 747. Each of these 12 clans was also represented in the genome of the streptophyte alga *K. nitens*. This indicates that these clans existed before the divergence of *K. nitens* and land plants from their last common ancestor.

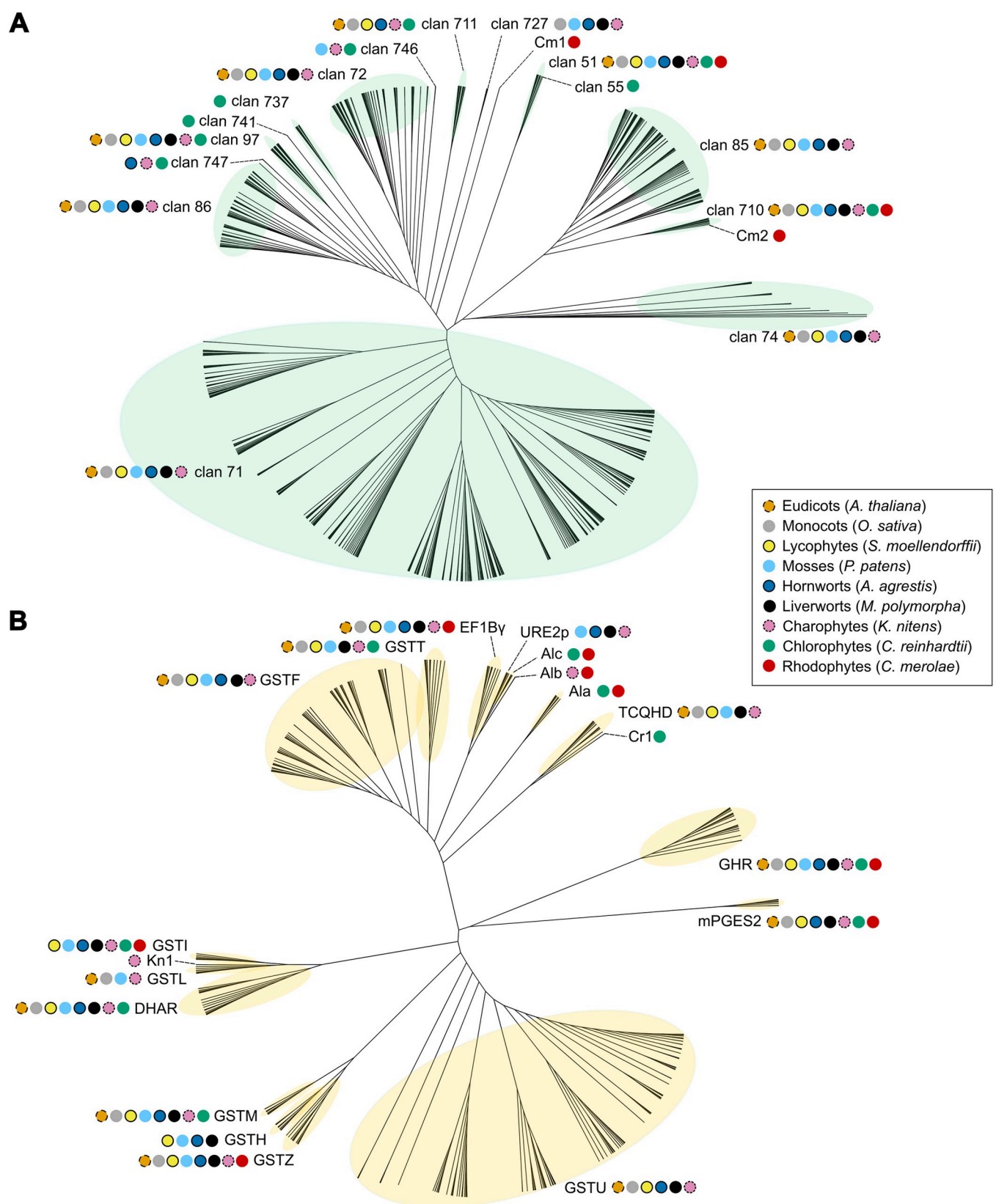

**Fig 1. Phylogenetic analysis of CYP and GST protein sequences in the Archaeplastida.** Unrooted cladogram of a maximum likelihood (ML) analysis of Archaeplastida CYP (A) and GST (B) proteins conducted by PhyML 3.0 [75] using an estimated gamma distribution parameter, the LG+G+F model of amino acid substitution, and a Chi$^2$-based approximate likelihood ratio (aLRT) test. Protein sequences were aligned using MAFFT with the FFT-NS-2 algorithm. CYP clans are indicated by light green highlighting and numbers. GST classes are indicated by light yellow highlighting and acronyms. Coloured dots indicate the presence of sequences from different species in each clan. *Arabidopsis thaliana* (orange); *Oryza sativa ssp. japonica* (grey); *Selaginella moellendorffii* (yellow); *Physcomitrium patens* (cyan); *Anthoceros agrestis* (blue); *Marchantia polymorpha* (black); *Klebsormidium nitens* (pink); *Chlamydomonas reinhardtii* (green); *Cyanidioschyzon merolae* (red). Charophytes is synonymous with streptophyte algae.

Members of 6 of the 12 clans– 71, 72, 74, 85, 86, and 727 –were not present in the genome of *C. reinhardtii* or in *C. merolae*. This suggests that these 6 clans originated in the streptophyte lineage after the divergence of chlorophytes and streptophytes from their last common ancestor but before the divergence of *K. nitens* (Fig 2A). Members of the other 6 of the 12 CYP clans– 51, 97, 710, 711, 746, and 747 –were encoded by the *C. reinhardtii* genome indicating that they were present before the divergence of streptophytes and chlorophytic algae from the last common ancestor. Two of the clans were also present in red algae; there is one member of clan 51 and two members of clan 710 in the genome of *C. merolae*. This places the origin of clan 51 and clan 710 before the divergence of Rhodophyta (red algae) and Viridiplantae (green plants) (Fig 2A). We conclude that clans 51 and 710 were present in the last common ancestor of Archaeplastida and therefore constitute the most ancient Archaeplastida clans.

Three clans– 55, 737, and 741 –were restricted to *C. reinhardtii*. There is a single clan 55 member in *C. reinhardtii*, CrCYP55B1, which was sister to the clan 51 clade. Members of clan 55 are also present in fungi and are hypothesised by [83] to have been acquired by *C. reinhardtii* from fungi through horizontal gene transfer. Two *C. reinhardtii* CYP protein sequences– CrCYP741A1 and CrCYP768A1 –formed a monophyletic clade, clan 741, that was sister to the clade comprising clans 86, 97, and 747. Thirty *C. reinhardtii* CYP sequences formed a monophyletic clade–clan 737 –which was sister to the clade containing the 86, 97, 741, and 747 clans. These data are consistent with our hypothesis that clans 737 and 741 are chlorophyte specific.

Two clans–Cm1 and Cm2 –comprised only single red algae proteins. Cm1 (CMD096C) was sister to the clade containing clans 72, 86, 97, 711, 727, 737, 741, 746, and 747. Clan Cm1 and clans 72, 86, 97, 711, 727, 737, 741, 746, and 747 are therefore likely derived from a protein present in the common ancestor of the red algae and the green plant lineage (chlorophytes and streptophytes). Cm2 (CMR093C) was sister to clan 710 but shares very low amino acid identity (20%) with members of 710. Cm2 is possibly an ancestral 710 protein or it could represent a red-algae specific clan. Clans Cm2 and 710 are therefore likely derived from a protein present in the common ancestor of the red algae and the green plant lineage (chlorophytes and streptophytes).

In summary, our phylogenetic analysis shows that each of the land plant CYP clans are also present in the genome of the streptophyte alga *K. nitens*. This indicates that the diversity of CYP sequences in plants evolved among algae in the aquatic environment before plants colonised land between 980 and 470 Mya [52–54]. Our analysis also shows that no new clans evolved among land plants after their colonisation of the land. Instead, the number of genes in each clan increased. Four CYP clans– 97, 711, 746 and 747—present in land plants and streptophyte algae are also present in the genome of the chlorophyte alga *C. reinhardtii*, which places their origin before the divergence of the chlorophyte and streptophyte lineages from their last common ancestor. Two clans found in land plants, streptophyte algae, and chlorophytes– 51 and 710 –are also present in the red algae. This suggests that these clans are the most ancient Archaeplastida clans and evolved before the divergence of Rhodophyta and Viridiplantae from their last common ancestor.

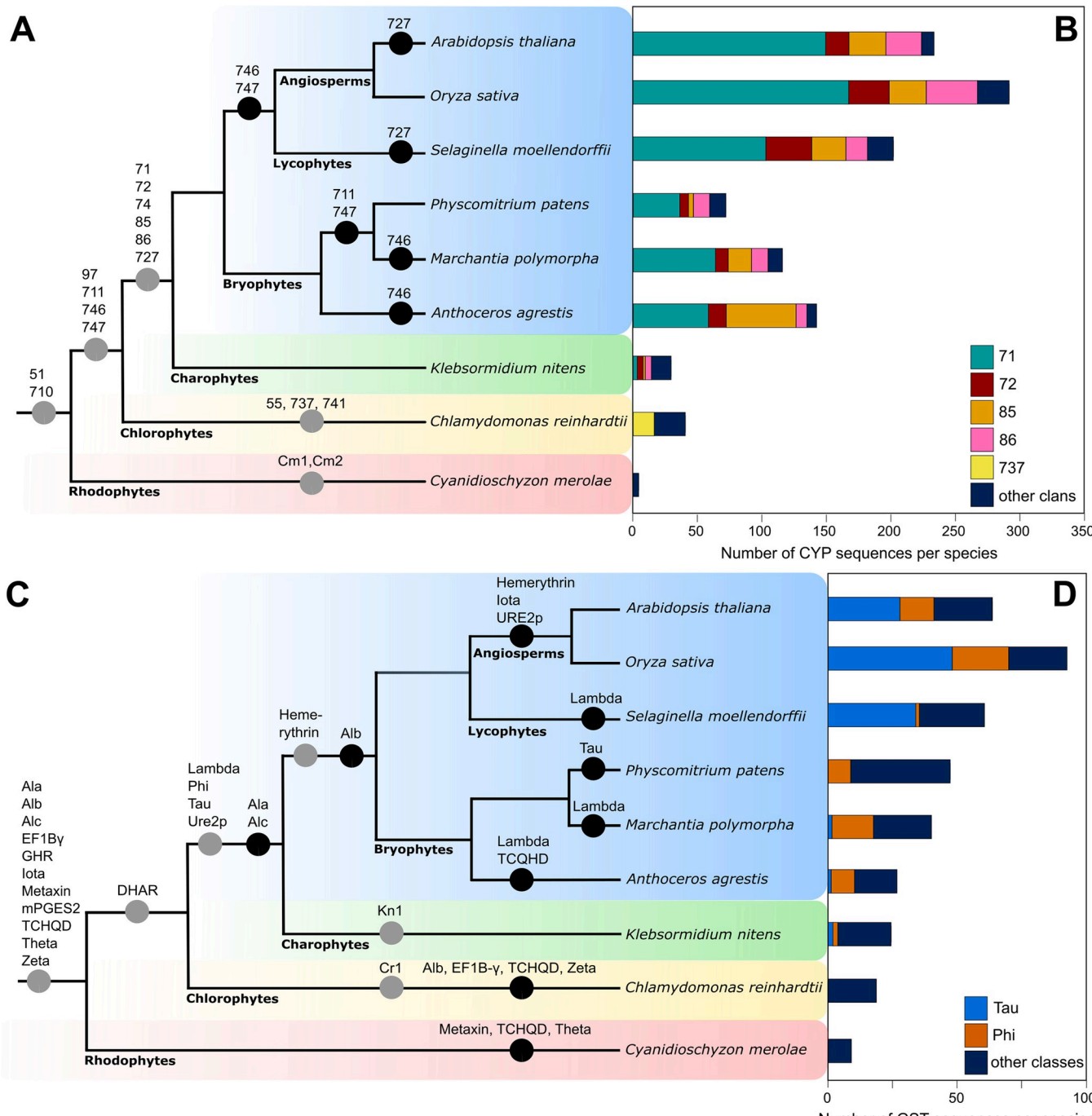

**Fig 2. Four CYP clans and two GST classes expanded during land plant evolution.** Cladogram of Archaeplastida phylogeny showing CYP clan (A) and GST class (C) origins and losses in plants. Grey circles represent first appearance of a clan or class, black circles represent the absence of a clan or class in a particular lineage. Numbers of CYP proteins in each species showing increases in the sizes of four of the five CYP clans shown (B) and of two GST classes (D) during land plant evolution.

## Plant GST classes are ancient, and 11 classes existed in the last common ancestor of the Archaeplastida

To elucidate the evolutionary history of GST classes in Archaeplastida, sequences were retrieved, aligned, and a phylogenetic tree constructed using maximum likelihood statistics (Fig 1B). The topology of the trees demonstrated that GSTs from the 9 representative species of Archaeplastida constituted 19 monophyletic classes–Ala, Alb, Alc, Cr1, DHAR, EF1B-γ, GHR, hemerythrin, iota, Kn1, lambda, metaxin, mPGES2, phi, tau, TCHQD, theta, Ure2p, and zeta. Of these 19 classes, 14 are encoded in the genomes of the land plant species *A. agrestis*, *M. polymorpha*, *P. patens*, S. *moellendorffii*, *O. sativa* and *A. thaliana*–DHAR, EF1B-γ, GHR, hemerythrin, iota, lambda, metaxin, mPGES2, phi, tau, TCHQD, theta, Ure2p and zeta (Fig 1B). Five of the 19 classes are novel GST classes identified in algal genomes, named Ala, Alb, Alc, Cr1, and Kn1.

Sixteen algal GST sequences comprised several different monophyletic clades. Three *C. reinhardtii* sequences and one *C. merolae* sequence comprised class Alc, which is a sister to the Ure2p class (Fig 1B). However, these sequences lacked a characteristic Ure2p protein domain (cd03048) and were therefore not included in the Ure2p class. Class Alb, which included one *K. nitens* sequence and one *C. merolae* sequence, is a sister to the monophyletic clade comprising both the Ure2p and Alc classes. Class Ala, comprising 7 *C. reinhardtii* sequences and a single *C. merolae* sequence, is a sister to the clade containing phi, theta, EFB1-γ, Ure2p, Alb, and Alc GST sequences. Ala, Alb, and Alc may represent classes that evolved in the ancestor of Archaeplastida, where Ala and Alc were lost in the common ancestor of streptophytes, and Alb was lost in the chlorophyte lineage and in the common ancestor of land plants (Fig 2C).

Two individual algal sequences formed two independent clades. A *C. reinhardtii* sequence (Cre12.g508850.t1) was sister to the TCHQD class (Fig 1B). However, this sequence lacked a TCHQD protein domain (IPR044617) and was therefore designated Cr1. A *K. nitens* sequence (Kfl00304_0120_v1) was sister to the lambda class (Fig 1B), however there was no GST lambda class C-terminal domain (cd03203). This sequence was designated Kn1. These data suggest that Cr1 evolved in the chlorophyte lineage and Kn1 evolved in the streptophyte algal lineage (Fig 2C).

Of the 14 GST classes present in the genomes of the land plants *A. agrestis*, *M. polymorpha*, *P. patens*, S. *moellendorffii*, *O. sativa* and *A. thaliana*, 9 classes–EF1B-γ, GHR, metaxin, mPGES2, phi, TCHQD, theta, Ure2p, zeta–are also found in non-plant genomes (such as metazoans, bacteria, archaea, and fungi) and therefore predate the origin of the Archaeplastida [27–30,32,33]. The other 5 GST classes–DHAR, hemerythrin, iota, lambda, and tau–have previously been described in the genomes of land plants and chlorophyte and streptophyte algae [28,65,67,84]. Our analysis shows that lambda and tau members are present in the genome of the streptophyte alga *K. nitens* but not in the *C. reinhardtii* and *C. merolae* genomes. This indicates that these classes evolved among the streptophytes after the divergence of the red algae and chlorophytes but before the divergence of *K. nitens* and land plants. Members of the hemerythrin class were found in genomes of the bryophytes (non-vascular plants) *P. patens*, *M. polymorpha*, and *A. agrestis* and the lycophyte S. *moellendorffii*, but not in the angiosperms or in *K. nitens*, *C. reinhardtii*, or *C. merolae*. This suggests that the hemerythrin class originated in the common ancestor of bryophytes and vascular plants but was lost in the common ancestor of the angiosperms. There are DHAR members in the genomes of *K. nitens* and *C. reinhardtii*. This suggests that DHAR GST proteins were present in the last common ancestor of chlorophytes and streptophytes. There are iota members in *C. merolae*, *C. reinhardtii*, and *K. nitens* indicating that iota class enzymes originated before the divergence of rhodophytes and chlorophytes in the common ancestor of Archaeplastida (Fig 2C).

There are 26 GST proteins belonging to 12 classes in the genome of the hornwort *Anthoceros agrestis* (S2 Table). One sequence (AagrOXF_evm.model.utg000005l.356.1) nested within the monophyletic tau GST clade and contained the conserved N- and C-terminal Tau class catalytic motifs (cd03058 and cd03185). This is strong evidence that AagrOXF_evm. model.utg000005l.356.1 is a tau class GST. Tau GST proteins are also present in streptophyte algae, liverworts, and vascular plants but absent from mosses. This suggests that the tau GST class was present in the last common ancestor of the streptophyte algae and subsequently lost in the moss lineage (Fig 2C).

In summary, this analysis showed that Archaeplastida GST proteins comprise 19 classes. 11 classes–Ala, Alb, Alc, EF1B-γ, GHR, iota, metaxin, mPGES2, TCHQD, theta, and zeta–were present in the common ancestor of the Archaeplastida. Of these, EF1B-γ, GHR, metaxin, mPGES2, TCHQD, theta, Ure2p and zeta are found in non-Archaeplastida genomes and therefore evolved before the divergence of the Archaeplastida [27–30,32,33]. Twelve classes originated after the divergence of Archaeplastida from other eukaryotes. The earliest GST classes to arise in Archaeplastida were the Ala, Alb, Alc, and iota classes, which originated before the separation of rhodophyte and chlorophyte lineages. The DHAR class originated in the common ancestor of chlorophytes and streptophytes. The Cr1 class originated in the chlorophyte lineage. Lambda, tau, phi, and Ure2p GSTs originated in the last common ancestor of streptophyte algae and land plants. Kn1 originated in the streptophyte algae. The most recently diverging plant GST class, the hemerythrin class, originated in the last common ancestor of land plants.

## CYP clans 71, 72, 85, and 86 and GST classes phi and tau GST expanded among land plants

The number of CYP genes encoded in the genomes of land plants is larger than the number encoded in the genomes of algae. We identified between 5 and 40 CYP protein genes in algae– 5 in *C. merolae*, 40 in *C. reinhardtii*, and 29 in *K. nitens*. We identified between 69 and 144 among the bryophytes– 69 in *A. agrestis*, 115 in *P. patens*, and 144 in *M. polymorpha* genomes. Among the vascular plants we identified between 199 and 291–199 in *S. moellendorffii*, 238 in *A. thaliana*, and 291 in *O. sativa* genomes (Tables 1 and S1).

To determine if CYP gene numbers are correlated with the numbers of total protein coding genes in land plants, we calculated the percentage of protein-coding genes that encoded CYP proteins. CYPs represent 0.18% of the protein-coding genes in the streptophyte alga *K. nitens*, 0.19–0.60% in bryophytes, and 0.82–0.93% in vascular plants (Table 1). These data are consistent with the hypothesis that the larger numbers of CYP genes in bryophytes and vascular plants than in algae are the result of CYP family expansion in land plants.

To identify the clans responsible for the higher proportion of protein-coding genes encoding CYPs in land plants than in algae, clan gene numbers were compared between species. There are more genes in clans 71, 72, 85, and 86 in land plants than in streptophyte algae (Fig 2B, S1 Table), with clan 71 gene numbers differing the most between species. There are three 71 clan members in the genome of the streptophyte alga *K. nitens*. Among the bryophytes there are 59 clan 71 members in the hornwort *A. agrestis*, 68 in the liverwort *M. polymorpha* and 38 in the moss *P. patens*. Among the vascular plants there are 98 in the lycophyte *S. moellendorffii*, 148 in *A. thaliana*, and 163 in *O. sativa* (S1 Table). Clan 71 proteins represent 10% of all CYPs in *K. nitens* but 40–60% of all CYPs in the land plants. Together these data are consistent with our hypothesis that the expansion in the numbers of clan 71 genes contributed to the large number of CYP proteins in land plants compared to algae (non-land plant Archaeplastida). There are only a small number of genes in eight CYP clans across all streptophyte species– 51, 74, 97, 710, 711, 727, 746, and 747. Generally, there were fewer than 10 members

in each of these clans in any one species (S1 Table). Thus, these clans therefore represent monophyletic groups that did not diversify among land plants.

Despite the smaller number of GST classes in land plants compared to algae, there are more GST protein coding genes in land plants than in algae. We identified 9 GST genes in the genome of *C. merolae*, 19 in *C. reinhardtii*, and 24 in *K. nitens*. Among the bryophytes we identified 35 in *M. polymorpha*, 42 in *P. patens* and 26 in *A. agrestis*. Among the vascular plants we identified 57 in *S. moellendorffii*, 85 in *O. sativa* and 61 in *A. thaliana* (Tables 1 and S2). Genes coding for GST proteins represent 0.15% of all protein coding genes in *K. nitens*, 0.11–0.18% in bryophytes, and 0.24–0.26% in vascular plants (Table 1). These data are consistent with our hypothesis that the larger number of GST genes in vascular plants than in algae is not because of a greater total number of protein-coding genes, but because of GST family expansion.

To identify the classes responsible for the increase in GSTs in vascular plants, gene numbers in each GST class were compared between species. The number of GST proteins in the phi and tau classes is larger in land plants than in streptophyte algae. There are 3 phi class members in the genome of the streptophyte alga *K. nitens*. Among the bryophytes there are 18 phi class genes in the genome of *M. polymorpha*, 10 in *P. patens* and 11 in *A. agrestis*. Only 1 phi GST was identified in the genome of the lycophyte *S. moellendorffii*. Among the angiosperms, there are 19 phi GST proteins in *O. sativa* and 13 in *A. thaliana*. This suggests that the phi class expanded in the land plant lineage after the divergence of streptophyte algae and land plants from the last common ancestor but before the divergence of bryophytes and vascular plants The identification of a single phi GST in *S. moellendorffii* suggests that phi class genes were lost in the lycophyte lineage. There are also more tau class GST proteins in vascular plant genomes than in either the algal or bryophyte genomes (Fig 2D). There are 3 tau class genes in the genome of *K. nitens*. Among the early diverging land plants there are 2 tau class members in *M. polymorpha*, 1 in *A. agrestis* and none in *P. patens*. Among the vascular plants there are 34 in *S. moellendorffii*, 49 in *O. sativa* and 28 in *A. thaliana*. This suggests that the tau class expanded in vascular plants after the divergence of bryophytes and vascular plants. In the other 17 GST classes in Archaeplastida–Ala, Alb, Alc, Cr1, DHAR, EF1B-γ, GHR, hemerythrin, iota, Kn1, lambda, metaxin, mPGES2, TCHQD, theta, Ure2p, and zeta–gene numbers are less than 10 in each species (S2 Table), indicating that these classes have not expanded during the course of evolution.

To further understand the pattern of tau and phi class expansions in land plants we compared the ratio of tau to phi GST proteins in each species. The tau/phi ratio in the streptophyte alga *K. nitens* is 1 (3 tau and 3 phi proteins). The ratio is less than 1 in the bryophyte genomes– 0.09 *A. agrestis* (1 tau, 11 phi proteins), 0 in *P. patens* (0 tau and 1 phi protein) and 0.11 in *M. polymorpha* (2 tau and 18 phi proteins) indicating that the phi class expanded more than the tau class in these species. The ratio is greater than 1 in the vascular plant genomes– 34 in *S. moellendorffii* (34 tau and 1 phi proteins), 5.57 in *O. sativa* (49 tau and 19 phi proteins) and 2.15 in *A. thaliana* (28 tau and 13 phi proteins)–indicating that the tau class expanded more than the phi class in these species.

In summary, our phylogenetic analysis shows that the 2 to 10-fold larger number of CYP genes in the genomes of land plants than in the streptophyte alga *K. nitens* results from expansions of clans 71, 72, 85, and 86. The 1.5 to 3.5-fold more GST genes in land plants than in the streptophyte alga *K. nitens* results from expansions of the phi and tau classes.

## Herbicide resistance and tolerance are associated with proteins from the GST phi and tau classes and CYP 71 and 72 clans

GSTs and CYPs have been genetically associated with herbicide resistance or tolerance in crop and weed species [85,86]. To identify which CYP clans and GST classes are genetically and/or

metabolically associated with herbicide resistance, a literature search was conducted. CYPs or GSTs reported in previous studies to increase herbicide resistance in transgenic plants or to metabolise herbicides were classified as NTSR genes (Tables 2 and 3). CYPs and GSTs found to have increased expression in herbicide resistant weeds, but whose function was not experimentally validated, were classified as "candidate NTSR genes" and are listed in S3 and S4 Tables.

## Clan 71 and clan 72 CYP proteins are associated with resistance or tolerance to herbicides from 18 chemical classes

A total of 30 plant CYPs have been experimentally shown to metabolise or confer resistance or tolerance to one or more herbicides in sensitivity or metabolism assays (Fig 3A, Table 2). These CYPs were identified in the model plant Arabidopsis (*A. thaliana*) [88,94], the grass weeds barnyard grass (*Echinochloa phyllopogon*) [99–101], shortawn foxtail (*Alopecurus aequalis*) [108] and annual ryegrass (*Lolium rigidum*) [102], the gymnosperm western red cedar (*Thuja plicata*) [92], and the crops barley (*Hordeum vulgare*) [103], rice (*Oryza sativa*) [95,106,107], wheat (*Triticum aestivum*) [90], maize (*Zea mays*) [96], cotton (*Gossypium hirsutum*) [109], soybean (*Glycine max*) [89,104], ginseng (*Panax ginseng*) [105], Jerusalem artichoke (*Helianthus tuberosus*) [91,93] and tobacco (*Nicotiana tabacum*) [87]. These 30 CYPs metabolised or conferred resistance or tolerance to diverse herbicide chemical classes, with the majority (25 of 30) metabolising phenylureas or sulfonylureas (Table 2).

All 30 of the herbicide-metabolising CYPs belong to clan 71 or 72 (Fig 3A). Twenty-six clan 71 enzymes have been shown to confer resistance to aryl-carboxylates (HRAC code 19), benzoate (HRAC code 4), benzothiadiazinone (HRAC code 6), isoxazolidinone (HRAC code 13), N-phenyl-triazolinone (HRAC code 14), phenylpyrazoline (DEN) (HRAC code 1), cyclohexanedione (DIM) (HRAC code 1), aryloxyphenoxypropionate (FOP) (HRAC code 1), phenylurea (HRAC code 5), pyrazole (HRAC code 27), pyridazinone (HRAC code 5), pyrimidinyl benzoate (HRAC code 2), sulfonylurea (HRAC code 2), thiadiazine (HRAC code 6), triazolinone (HRAC code 2), triazolopyrimidine (HRAC code 2) and triketone (HRAC code 27) herbicide chemicals [133]. Clan 71 CYPs are encoded in large number in the genomes of all land species; there are 150 in *A. thaliana* and 164 in *O. sativa*. In contrast, there are much fewer clan 72 CYPs encoded in land plant genomes, with 19 in *A. thaliana* and 34 in *O. sativa*. Four clan 72 members were shown to confer resistance to pyrimidinyl benzoates (HRAC code 2), pelargonic acid (HRAC code 0—other), or sulfonylureas (HRAC code 2) [133]. Thus, all CYPs currently known to metabolise or confer resistance to herbicides belong to clans 71 and 72, which represent two of the four expanded CYP clans in land plants.

Twelve members of the clan 71 family CYP81 were shown to confer herbicide resistance. This is more than any other family or clan (Fig 3C). The CYP81 enzymes metabolise herbicides from five chemical classes, more than any other CYP family. CYP81 enzymes catalyse hydroxylations and N-/O-demethylations of herbicide substrates [99]. Together these data indicate that genes encoding CYP proteins that confer herbicide resistance or tolerance are members of clan 71 and 72. Within clan 71, more members of the CYP81 family confer herbicide resistance or tolerance than any other family.

## Phi, tau and lambda GST class proteins are associated with resistance or tolerance to herbicides from 9 chemical classes

Thirty-three plant GSTs were found in the literature to be active towards one or more herbicides or that confer herbicide resistance (Fig 3B, Table 3). These GST proteins were identified in the model species Arabidopsis (*Arabidopsis thaliana*) [125], moss (*P. patens*) [28], the weed

**Table 2. Plant CYPs that metabolise or confer resistance or tolerance to herbicides are found within clans 71 and 72.** Table adapted from [11]. Gene numbers from this table are shown in Fig 3.

| Clan | Sub-family | Gene name and species | Evidence | Herbicide class | Genes per clan | References |
|---|---|---|---|---|---|---|
| 71 | 71A | NtCYP71A11 (tobacco) | Metabolism in yeast | Phenylurea | 26 | [87] |
| | 71A | AtCYP71A12 (Arabidopsis) | Metabolism in yeast | Pyrazole | | [88] |
| | 71A | GmCYP71A10 (soybean) | Transformation in tobacco | Phenylurea | | [89] |
| | 71C | TaCYP71C6v1 (wheat) | Metabolism in yeast | Sulfonylurea | | [90] |
| | 73A | HtCYP73A1 (Jerusalem artichoke) | Metabolism in yeast | Phenylurea | | [91] |
| | 76A | TpCYP76AA20 (western redcedar) | Metabolism in yeast | Phenylurea | | [92] |
| | 76A | TpCYP76AA21 (western redcedar) | Metabolism in yeast | Phenylurea | | [92] |
| | 76A | TpCYP76AA22 (western redcedar) | Metabolism in yeast | Phenylurea | | [92] |
| | 76A | TpCYP76AA25 (western redcedar) | Metabolism in yeast | Phenylurea | | [92] |
| | 76B | HtCYP76B1 (Jerusalem artichoke) | Metabolism in yeast | Phenylurea | | [93] |
| | 76C | AtCYP76C1 (Arabidopsis) | Metabolism in yeast | Phenylurea | | [94] |
| | 76C | AtCYP76C2 (Arabidopsis) | Metabolism in yeast | Phenylurea | | [94] |
| | 76C | AtCYP76C4 (Arabidopsis) | Metabolism in yeast | Phenylurea | | [94] |
| | 81A | OsCYP81A6 (rice) | Knock-out in rice | Thiadiazine, Sulfonylurea | | [95] |
| | 81A | ZmCYP81A9 (maize) | Gene-silencing in maize | Aryl-carboxylate, Benzoate, Benzothiadiazinone, N-phenyl-triazolinone, Sulfonylurea, Triketone | | [96–98] |
| | 81A | EcCYP81A12 (barnyard grass) | Transformation in *A. thaliana*, *E. coli* and yeast | Benzothiadiazinone, DEN, DIM, Pyrimidinyl benzoate, Triazolinone, Sulfonylurea, Triazolopyrimidine, Triketone | | [99,100] |
| | 81A | EcCYP81A14 (barnyard grass) | Transformation in *A. thaliana*, metabolism in *E. coli* | Pyrimidinyl benzoate, Triazolinone, Sulfonylurea | | [99] |
| | 81A | EcCYP81A15 (barnyard grass) | Transformation in *A. thaliana*, rice, metabolism in *E. coli* | Benzothiadiazinone, Isoxazolidinone, DEN, DIM, Pyrimidinyl benzoate, Sulfonylurea | | [99,101] |
| | 81A | EcCYP81A18 (barnyard grass) | Transformation in *A. thaliana*, metabolism in *E. coli* | Pyrimidinyl benzoate, Triazolinone, Sulfonylurea | | [99] |
| | 81A | EcCYP81A21 (barnyard grass) | Transformation in *A. thaliana*, metabolism in *E. coli* & yeast | Benzothiadiazinone, Isoxazolidinone, Pyrimidinyl benzoate, Triazolinone, Sulfonylurea, Triazolopyrimidine | | [99,100] |
| | 81A | EcCYP81A24 (barnyard grass) | Transformation in *A. thaliana*, rice, metabolism in *E. coli* | Benzothiadiazinone, Isoxazolidinone, Pyrazole, Pyridazinone, Pyrimidinyl benzoate, Sulfonylurea, Triazolopyrimidine, Triketone | | [99,101] |
| | 81A | LmCYP81A10v7 (annual ryegrass) | Transformation and metabolism in rice | DIM, FOP, Sulfonylurea | | [102] |
| | 81A | HvCYP81A63 (barley) | Transformation in rice | DEN, DIM, FOP | | [103] |
| | 81B | NtCYP81B2 (tobacco) | Metabolism in tobacco cells | Phenylurea | | [87] |
| | 81E | GmCYP81E22 (soybean) | Transformation in soybean | Benzothiadiazinone | | [104] |
| | 736A | PgCYP736A12 (ginseng) | Overexpression in *A. thaliana* | Phenylurea | | [105] |

*(Continued)*

**Table 2.** (Continued)

| Clan | Sub-family | Gene name and species | Evidence | Herbicide class | Genes per clan | References |
|---|---|---|---|---|---|---|
| 72 | 72A | OsCYP72A31 (rice) | Overexpression in *A. thaliana* | Pyrimidinyl benzoate | 4 | [106] |
| | 72A | OsCYP72A18 (rice) | Metabolism in yeast microsomes | Pelargonic acid | | [107] |
| | 709C | AaCYP709C56 (shortawn foxtail) | Transformation in *A. thaliana*, metabolism in yeast | Sulfonylurea | | [108] |
| | 749A | GhCYP749A16 (cotton) | Gene-silencing in cotton | Sulfonylurea | | [109] |

species blackgrass (*Alopecurus myosuroides*) [123,124], the crops maize (*Zea mays*) [111–118,121,130,131], rice (*Oryza sativa*) [110,121,134], sorghum (*Sorghum bicolor*) [122], wheat (*Triticum aestivum*) [24,119,120,132] and soybean (*Glycine max*) [126–128]. These GSTs were shown to modify or confer resistance to diverse chemical classes, with most GSTs (28 of 33) modifying α-chloroacetamide (HRAC code 15) herbicides. Of the 33 GSTs, 11 are phi class members, 21 are tau class members, and one is a lambda class member (Fig 3B).

Twenty-one tau GSTs were identified in 6 species and catalysed the GSH-conjugation of α-chloroacetamide (HRAC code 15), diphenyl ether (HRAC code 14), FOP (HRAC code 1), sulfonylurea (HRAC code 2) and triazine (HRAC code 5) herbicide chemicals. Eleven phi GSTs identified in 6 species catalysed the GSH-conjugation of bipyridylium (HRAC code 22), α-chloroacetamide (HRAC code 15), DIM (HRAC code 1), diphenyl ether (HRAC code 14), FOP (HRAC code 1), glycine (HRAC code 9), phenylurea (HRAC code 5), sulfonylurea (HRAC code 2), thiocarbamate (HRAC code 15) and triazine herbicides (HRAC code 5) (Table 3) [133].

The majority of GSTs encoded in the genomes of vascular plants *S. moellendorffii* (61%), *O. sativa* (80%) and *A. thaliana* (67%) are tau or phi class members. In *A. thaliana*, there are 41 tau and phi GSTs and 20 GSTs across the other 12 classes. In *O. sativa*, there are 68 tau and phi GSTs and 17 in the other classes (Fig 3B). Thus, the overrepresentation of phi and tau class GSTs among those reported to confer herbicide resistance may simply be due to the fact that there are more genes in these classes than others. Therefore, we cannot reject the hypothesis presented in this paper that there is an equal probability of GST proteins from any class being able to confer herbicide resistance. The report that overexpression of a single lambda class GST–there are 3 lambda class genes encoded in *A. thaliana*–can confer herbicide resistance supports this hypothesis.

## Discussion

Cytochrome P450 monooxygenases (CYPs) and glutathione S-transferases (GSTs) are enzymes that catalyse the metabolism of a multitude of organic compounds in organisms from all domains of life. Overexpression of genes encoding CYPs and GSTs has been shown to confer herbicide resistance in wild weed populations subjected to herbicide selection. To classify the genes that metabolise herbicides, we carried out a phylogenetic analysis of both the CYP and GST protein families. By comparing protein sequences of 9 representative species of the Archaeplastida–the lineage that includes the red algae, glaucophyte algae, chlorophyte algae, and streptophytes–and generating phylogenetic trees, we identified that members of two CYP clans (clans 51 and 710) and eleven GST classes (Ala, Alb, Alc, EF1B-y, GHR, iota, metaxin, mPGES2, TCHQD, theta, and zeta) existed in the last common ancestor of the Archaeplastida. Other clans and classes evolved over the course of Archaeplastida evolution. There are more CYP and GST genes in land plants than in algae, even relative to the total number of genes, consistent with our hypothesis that these gene families expanded during Archaeplastida

**Table 3. Plant GST proteins that conjugate or confer resistance or tolerance to herbicides are members of classes phi, tau, and lambda.** *O. sativa* and *A. thaliana* genes were renamed according to current nomenclature [22]. Gene numbers from this table are shown in Fig 3.

| Class | Gene name | Evidence | Herbicide chemical class | Genes per class | References |
|-------|-----------|----------|--------------------------|-----------------|------------|
| Lambda | OsGSTL1 (rice) | Overexpression in rice | Glycine and Sulfonylurea | 1 | [110] |
| Phi | ZmGSTF1 (maize) | Conjugating activity in vitro, overexpressed in tobacco | α-Chloroacetamide | 13 | [111–113] |
| | ZmGSTF2 (maize) | Conjugating activity in vitro, transformed in tobacco & wheat | α-Chloroacetamide, Diphenyl ether, Thiocarbamate | | [114–116] |
| | ZmGSTF3 (maize) | Conjugating activity in vitro & in *E. coli* | α-Chloroacetamide and Diphenyl ether | | [115,117] |
| | ZmGSTF4 (maize) | Conjugating activity in vitro, Transformed in tobacco | α-Chloroacetamide and Thiocarbamate | | [116,118] |
| | TaGSTF2-2 (wheat) | Conjugating and peroxidase activity in *E. coli* | α-Chloroacetamide and Diphenyl ether | | [119] |
| | TaGSTF3-3 (wheat) | Conjugating & peroxidase activity in *E. coli* | α -chloroacetamide and Diphenyl ether | | [119] |
| | TaGST2-3 (wheat) | Conjugating activity in vitro | Diphenyl ether | | [120] |
| | OsGSTF3-3 (rice) | Conjugating activity in *E. coli* | α-Chloroacetamide | | [121] |
| | SbGSTB1-B2 (sorghum) | Conjugating activity in vitro | α-Chloroacetamide | | [122] |
| | PpGSTF7 (moss) | Conjugating activity in *E. coli* | Diphenyl ether | | [28] |
| | AmGSTF1 (blackgrass) | Peroxidase activity in *E. coli*, regulation of flavonoids, transformation in *A. thaliana* | α-Chloroacetamide, Diphenyl ether, Triazine, Phenylurea, Bipyridylium | | [123,124] |
| Tau | AmGSTU1 (blackgrass) | Conjugating activity in *E. coli* | Diphenyl ether and FOP | 22 | [123] |
| | AtGSTU19 (Arabidopsis) | Conjugating activity in *E. coli* | α-Chloroacetamide | | [125] |
| | GmGSTU2 (soybean) | Conjugating activity in *E. coli* | Triazine | | [126] |
| | GmGSTU4 (soybean) | Conjugating and peroxidase activity in *E. coli*, overexpressed in tobacco | α-Chloroacetamide and Diphenyl ether | | [126,127] |
| | GmGSTU5 (soybean) | Conjugating activity in *E. coli* | α-Chloroacetamide and Sulfonylurea | | [126] |
| | GmGSTU7 (soybean) | Conjugating activity in *E. coli* | Triazine | | [126] |
| | GmGSTU8 (soybean) | Conjugating activity in *E. coli* | α-Chloroacetamide | | [126] |
| | GmGSTU9 (soybean) | Conjugating activity in *E. coli* | α-Chloroacetamide and Sulfonylurea | | [126] |
| | GmGSTU10 (soybean) | Conjugating activity in *E. coli* | α-Chloroacetamide and Sulfonylurea | | [126] |
| | GmGSTU21 (soybean) | Conjugating activity in *E. coli*, transformed in tobacco | Diphenyl ether | | [128] |
| | OsGSTU3 (rice) | Conjugating activity in *E. coli* | α-Chloroacetamide | | [129] |
| | OsGSTU4 (rice) | Conjugating activity in *E. coli* | α-Chloroacetamide | | [129] |
| | ZmGSTU1 (maize) | Conjugating activity in vitro & in *E. coli* | α-Chloroacetamide and Diphenyl ether | | [126,130] |
| | ZmGSTU2 (maize) | Conjugating activity in *E. coli* | α-Chloroacetamide and Diphenyl ether | | [131] |
| | ZmGSTU3 (maize) | Conjugating activity in *E. coli* | α-Chloroacetamide | | [131] |
| | ZmGSTU4/19 (maize) | Conjugating activity in *E. coli* | α-Chloroacetamide and Sulfonylurea | | [126] |

*(Continued)*

**Table 3.** (Continued)

| Class | Gene name | Evidence | Herbicide chemical class | Genes per class | References |
|-------|-----------|----------|--------------------------|-----------------|------------|
| | TaGSTU1-1 (wheat) | Conjugating and peroxidase activity in vitro | α-Chloroacetamide, Diphenyl ether | | [132] |
| | TaGSTU1-2 (wheat) | Conjugating and peroxidase activity in vitro | α-Chloroacetamide, Diphenyl ether, FOP, Triazine | | [132] |
| | TaGSTU1-3 (wheat) | Conjugating activity in vitro | α-Chloroacetamide, Diphenyl ether, FOP, Triazine | | [132] |
| | TaGSTU1-4 (wheat) | Conjugating and peroxidase activity in vitro | α-Chloroacetamide, Diphenyl ether, FOP | | [132] |
| | TaGSTU4-4 (wheat) | Conjugating activity in vitro & in *E. coli* | α-Chloroacetamide and FOP | | [24] |

evolution. Our analyses indicate that this expansion was largely driven by gene duplications among CYP clans 71 and 72, and among the GST phi and tau classes [1,2]. The ratio of tau to phi GSTs varies in different land plant lineages. There are more phi GSTs than GSTs in bryophyte genomes, while there are more tau GSTs than phi GSTs in vascular plant genomes.

In the face of intense herbicide use over the past 50 years, herbicide resistance has evolved through the selection of naturally occurring alleles that contribute to resistance. Genes encoding CYPs and GSTs are associated with herbicide resistance in many weed populations [9–11]. We show that the CYP and GST proteins that confer non-target site herbicide resistance in weed populations and tolerance in crop plants belong to the expanded CYP clans 71 and 72 and the GST phi and tau classes.

It is unclear why enzymes in CYP clans 71 and 72, and GST phi and tau classes metabolise herbicides while enzymes in other clans and classes do not. It is possible that because these clans and classes are the largest, there is simply a greater probability of these proteins conferring resistance. It is also possible that the enzymatic activity of these proteins makes them more likely to metabolize herbicide compounds. Proteins from CYP clans 71 and 72 catalyse the oxidation of diverse substrates in plants including small molecules that are intermediates in biosynthetic pathways of hormone compounds like strigolactones, gibberellin, brassinosteroids, and a range of secondary metabolites such as flavonoids and the phytoalexins such as camalexin [135]. Phi GST enzymes catalyse reactions involved in anthocyanin transport, and the synthesis of diverse plant defence compounds [1,65,136–138]. Similarly, tau GSTs are involved in the synthesis of defence compounds, and catalyse reactions with diverse chemical classes such as porphyrin derivatives, anthocyanins, and fatty acids [139–141]. It is possible that the structures of the natural substrates for some of these enzymes may resemble structures of herbicides, making the latter susceptible to catalysis. However, we are not aware of such similarities between herbicides and the natural substrates of clan 71 and 72 CYP proteins, or the natural substrates of phi and tau GST proteins. One unifying feature of GST substrates is an electrophilic carbon that is directly conjugated by the thiolate anion of glutathione [142]. The majority of herbicides have an electrophilic centre, therefore this supports the hypothesis that all GSTs have the potential to metabolise electrophilic herbicides [142,143]. Further characterization of the endogenous function of CYP clans 71 and 72 and GST tau and phi classes during normal plant growth and development will help to answer this question. At present, the available phylogenetic and enzymatic data do not allow us to distinguish between these alternative hypotheses.

All CYP clans and all but one GST classes that are present in land plants evolved before the divergence of streptophyte algae and land plants from their last common ancestor. These

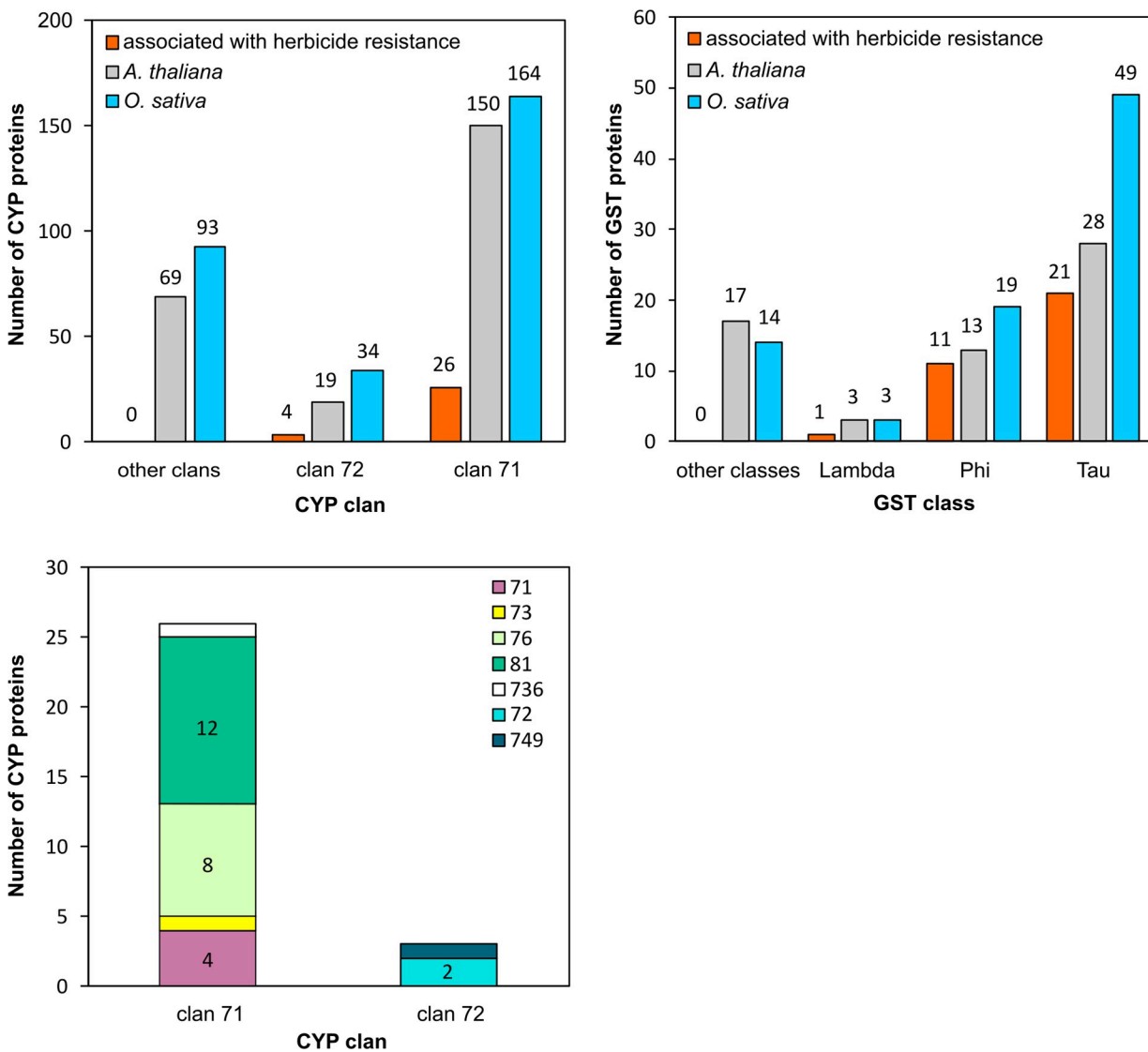

**Fig 3. GST and CYP proteins associated with herbicide resistance or tolerance belong to the lambda, phi, and tau classes and clans 71 and 72.** (A) Number of CYP proteins associated with herbicide resistance (orange bars), present in the *A. thaliana* genome (light grey bars) and in the *O. sativa* genome (blue bars), per clan. (B) Number of GST proteins associated with herbicide resistance (orange bars), present in the *A. thaliana* genome (light grey bars) and the *O. sativa* genome (blue bars) per class. (C) Number of CYP proteins associated with resistance per clan, with family membership indicated by colours. The most represented family among CYPs associated with herbicide resistance is the CYP81 family. Numbers over or within bars represent the number of proteins within that category.

results demonstrate that the clan and class diversity in extant plant CYP and GST proteins, respectively, evolved in the Proterozoic (before 538.8 Mya), before the divergence of land plants and streptophyte algae from a last common ancestor [144]. Then, early in embryophyte evolution during the Palaeozoic (251.9–538.8 Mya), expansion of four of the twelve CYP clans and two of the fourteen GST classes resulted in the large number of CYP and GST proteins found in extant land plants [144]. This expansion likely accompanied an increase in diversity of signalling molecules and secondary metabolites that may have occurred soon after plants started to grow in relatively dry terrestrial environments. It is among these expanded groups that herbicide resistance genes are found.

We showed that the major groups of CYP and GST genes evolved in the Proterozoic. Consequently, herbicide resistance evolved from changes in the activities of genes that evolved in the Proterozoic, whose original functions were unrelated to herbicide metabolism. The evolution of resistance through alteration of the function of these genes might be considered an example of exaptation. According to this model, gene variants that originally evolved with one function–probably metabolism–were selected to carry out an entirely different function–conferring herbicide resistance [145]. Exaptation is likely to be a general principle underpinning the evolution of herbicide resistance mechanisms among weeds in the agricultural landscape.

## Supporting information

**S1 Fig. Overview of cytochrome P450 and glutathione *S*-transferase protein features in plants.** Diagram of a typical CYP protein showing recognisable amino acid sites. GST G-site and H-site locations in this figure are based on the crystal structure of TaGSTU4.
(PDF)

**S2 Fig. Plant CYP and GST phylogenetic analysis using automatic and manual trimming approaches.** Unrooted cladograms of maximum likelihood (ML) analysis conducted by PHyML 3.0 [75] using an estimated gamma distribution parameter, the LG+G+F model of amino acid substitution and a Chi$^2$-based approximate likelihood ratio (aLRT) test. CYP (A) and GST (B) sequences were aligned in MAFFT and trimmed with the automatic trimming software trimAl using the automatic modes -strictplus, -strict, -gappyout or by manual trimming. Branches are coloured to show the different CYP clans or GST classes. aLRT Support values for some of the clades are shown for comparison.
(PDF)

**S3 Fig. Untrimmed amino acid alignment of representative CYP proteins from each clan showing the location of conserved CYP domains.** Representative sequences from each plant species in this study are included for each clan. Sequences were aligned in MAFFT using the FFT-NS-i algorithm. The locations of the substrate recognition sites are based on those identified in Arabidopsis CYPs in [37]. The absolutely conserved cysteine that binds the heme within the heme-binding domain is marked with an asterisk.
(PDF)

**S4 Fig. Amino acid alignment of representative plant GST proteins showing the location of conserved GST domains.** Sequences were aligned in MAFFT using the FFT-NS-i algorithm. Four representative sequences from different species are shown for each GST class. The location of the putative catalytic residue is indicated with an asterisk. Sites that bind GSH (G-sites) are indicated in solid pink. Residues conserved in at least 80% of samples are indicated by blue arrows. GSTHs and GSTIs have large domains that extend past the C-terminal domain end which haven't been included in the figure. G-site residues are based on the crystal structure of TaGSTU4 [24].
(PDF)

**S1 Table. Cytochrome P450 clans and gene numbers in green plants and red algae.** Numbers of CYP proteins in each clan, excluding pseudogenes. *At*, *Arabidopsis thaliana*; *Os*, *Oryza sativa*; *Sm*, *Selaginella moellendorffii*; *Aa*, *Anthoceros agrestis*; *Pp*, *Physcomitrium patens*; *Mp*, *Marchantia polymorpha*; *Kn*, *Klebsormidium nitens*; *Cr*, *Chlamydomonas reinhardtii*; *Cm*, *Cyanidioschyzon merolae*.
(PDF)

**S2 Table. Glutathione-S-transferase classes and gene numbers in green plants and red algae.** Numbers of GST proteins in each clan, excluding pseudogenes. *At*, *Arabidopsis*

*thaliana; Os, Oryza sativa; Sm, Selaginella moellendorffii; Aa, Anthoceros agrestis; Pp, Physcomitrium patens; Mp, Marchantia polymorpha; Kn, Klebsormidium nitens; Cr, Chlamydomonas reinhardtii; Cm, Cyanidioschyzon merolae.*
(PDF)

**S3 Table. Candidate NTSR CYPs belong to several CYP classes.**
(PDF)

**S4 Table. Candidate NTSR GSTs belong to several GST classes.**
(PDF)

**S5 Table. Number of GST proteins identified from classes 2N, Kappa, and MAPEG in green plants and red algae.** Sequences from these classes were not included in the phylogenetic analysis because they lack the classical N-terminal and C-terminal GST domains. 2N GST sequences have two N-terminal domains and lack a C-terminal domain. Kappa GST proteins lack both N and C-terminal GST domains and instead have a single thioredoxin-like kappa GST domain (InterPro domain IPR014440). MAPEG GST proteins lack both C and N-terminal GST domains and have instead a single 'MAPEG' domain (InterPro domain IPR001129).
(PDF)

**S1 Text. Untrimmed alignment of all CYP sequences used in the phylogenetic analysis.**
(TXT)

**S2 Text. Manually trimmed alignment of all CYP sequences used in the phylogenetic analysis.**
(TXT)

**S3 Text. Trimmed alignment of all CYP sequences used in the phylogenetic analysis using the trimAI -gappyout automated setting.**
(TXT)

**S4 Text. Trimmed alignment of all CYP sequences used in the phylogenetic analysis using the trimAI -strict automated setting.**
(TXT)

**S5 Text. Trimmed alignment of all CYP sequences used in the phylogenetic analysis using the trimAI -strictplus automated setting.**
(TXT)

**S6 Text. Untrimmed alignment of all GST sequences used in the phylogenetic analysis.**
(TXT)

**S7 Text. Manually trimmed alignment of all GST sequences used in the phylogenetic analysis.**
(TXT)

**S8 Text. Trimmed alignment of all GST sequences used in the phylogenetic analysis using the trimAI -gappyout automated setting.**
(TXT)

**S9 Text. Trimmed alignment of all GST sequences used in the phylogenetic analysis using the trimAI -strict automated setting.**
(TXT)

**S10 Text. Trimmed alignment of all GST sequences used in the phylogenetic analysis using the trimAI -strictplus automated setting.**
(TXT)

## Acknowledgments

The authors would like to thank Professor David Nelson for the nomenclature of *Marchantia polymorpha* CYPs in this article, and Dr Sandy Hetherington for his input and advice on the methods. We are grateful to Matt Watson for editorial advice and to Sarah Attrill, Chloe Casey, Sam Caygill, Hugh Mulvey and Shuangyang Wu for providing feedback on earlier drafts of this manuscript.

## Author Contributions

**Conceptualization:** Liam Dolan.

**Data curation:** Alexandra Casey.

**Formal analysis:** Alexandra Casey.

**Funding acquisition:** Liam Dolan.

**Investigation:** Alexandra Casey.

**Methodology:** Alexandra Casey.

**Project administration:** Alexandra Casey.

**Resources:** Liam Dolan.

**Software:** Alexandra Casey.

**Supervision:** Liam Dolan.

**Validation:** Alexandra Casey.

**Visualization:** Alexandra Casey.

**Writing – original draft:** Alexandra Casey.

**Writing – review & editing:** Liam Dolan.

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
