## [Decision Letter · Decision Letter 0]

2 Oct 2022

PONE-D-22-22418Genes encoding cytochrome P450 monooxygenases and glutathione S-transferases associated with herbicide resistance evolved before the origin of land plantsPLOS ONE

Dear Dr. Dolan,

Thank you for submitting your manuscript to PLOS ONE. After careful consideration, we believe the manuscript that has scientific merit but does not fully meet PLOS ONE’s publication criteria as it currently stands. Therefore, we invite you to submit a revised version of the manuscript that addresses the points raised during the review process.

 Summary of Reviewer & Editor Comments: 

- The manuscript reports interesting information regarding gene evolution of plant GSTs and CYPs, which I believe will be of great interest to plant biologists and crop protection scientists.

- No serious flaws were identified, but numerous minor points were raised by the three reviewers aimed at improving the clarity and accuracy of your manuscript.

- Most comments focus on eliminating redundancy in certain sections, tightening up the wording so the text flows better, and establishing a stronger link between endogenous enzyme functions vs. evolved NTSR mechanisms in weeds pertaining to xenobiotic metabolism. 

We look forward to receiving your revised manuscript.

Kind regards,

Dean E. Riechers, PhD

Academic Editor

PLOS ONE

Journal Requirements:

Reviewers' comments:

Reviewer's Responses to Questions

**Comments to the Author**

1. Is the manuscript technically sound, and do the data support the conclusions?

Reviewer #1: Yes

Reviewer #2: Yes

Reviewer #3: Yes

2. Has the statistical analysis been performed appropriately and rigorously? 

Reviewer #1: Yes

Reviewer #2: Yes

Reviewer #3: I Don't Know

3. Have the authors made all data underlying the findings in their manuscript fully available?

Reviewer #1: Yes

Reviewer #2: Yes

Reviewer #3: Yes

4. Is the manuscript presented in an intelligible fashion and written in standard English?

Reviewer #1: Yes

Reviewer #2: Yes

Reviewer #3: Yes

5. Review Comments to the Author

**Reviewer #1: General Comments:**

This manuscript is well-written, clear, and organized. The topic offers a new perspective on herbicide resistance making the subject of interest to weed scientists as well as those interested in molecular phylogenetics. The article is a unique blend a literature review of NTS-based herbicide resistance and the role specific CYPs and GSTs play in that resistance.

Specific Comments/Questions:

Throughout the manuscript there is mention of hypotheses and how the data support various hypotheses (lines 266, 271, 368, 378, 390, 486, 489, 507). This hypothesis-testing emphasis allows the reader to read the document with the purpose in mind. However, it is unclear as to whether the hypotheses mentioned are the authors’ or the disciplines. In my opinion, I would state the hypotheses being addressed and tested up front in the introduction so that the reader can read the article with a goal in mind.

One improvement to the document would be to add more information on the phylogeny of the species selected for the analysis. Table 1 partially serves this purpose. Maybe Table 1 could be modified to include how these vascular and non-vascular plants and alga fit into the kingdom. Also, terms like embryophyte, bryophyte, glaucophyte, and Viridiplante are used without context or definition. Adding this information would make the manuscript an easier read for agronomists and weed scientists.

Another piece of information that would be helpful is the rationale used to determine which species to include in this analysis. First, why were these classes selected? For example, are ferns and conifers included? Also, of the 9 species selected for the analysis, why these? Is it due to genomic information or genome size? Are they recognized representatives of their classes in the field of plant phylogeny?

The authors imply that TSR and NTSR of weeds is due to mutations (first paragraph of introduction; Lines 462-463; Line 524). This is not always true. For example, could an herbicide molecule and a native substrate for a CYP or GST both fit in the enzyme’s active site? Also, some NTSR cases of weed resistance may be due to the accumulation of alleles that, when expressed, together confer tolerance to an herbicide (i.e., multigenic). If the term mutation is being more broadly used to include gene duplication or changes in gene regulation, it should be made clear.

Finally, from the literature search and data analyses conducted, are the authors able to determine what percentage of NTSR cases are due to CYPs and/or GSTs? What percentage of metabolism-based resistance cases are not linked to any genes (unaccounted for)? Are there any linkages with the chemical classes subject to metabolism by GSTs and CYPs? In other words, are there any chemical motifs susceptible to metabolism by CYPs and GSTs? Determining the native function of these enzymes might enable this linkage.

Minor Suggestions/Corrections:

Introduction

Line 44: Does “inhibit the interaction between the two” mean “reduce affinity of the herbicide for the target site”?

Line 72: Is there a description for Ure2p?

Lines 70-74: Is there a reference for this statement/lists?

Line 93: A brief definition of Archaeplastida would be helpful here.

Line 96: It would be helpful to define MYA for its first use.

Line 99: Since this is the first mention of K. nitens in the document, the genus name should be used.

Materials & Methods

Line 116: Since this is the first mention of A. thaliana in the document, the genus name should be used. Also need a period at end of sentence.

Line 117: Was the ssp. japonica sequence used for Oryza sativa? If so, please indicate here. Figure 1 legend indicates the japonica sequence was used.

Line 118: Since this is the first mention of M. polymorpha in the document, the genus name should be used.

Lines 116-127: Why is the protein sequence reference for S. moellendorffii not included with the other?

Line 146: Maybe change “don’t” to “do not”.

Results

Line 255: Define Viridiplantae here if not added earlier in the manuscript.

Line 280: Phrase “of the” repeated in sentence.

Line 283: For consistency use MYA?

Line 284: Reference fig 2C?

Line 284: Should this be four CYPs instead of 5 CYPs?

Line 303: Start sentence with “Sixteen”.

Line 323: Iota is shown in the Rhodophytes

Line 347: Start sentence with “Twelve”.

Lines 348-350: Is there a reference for this statement? There is no information in the manuscript to differentiate these 4 classes from the other 7 that also are shown as the earliest.

Lines 399-401: How can these data account for there being only 1 Phi GST in S. moellendorffii? Please address.

Lines 427 & 448: “30” instead of “thirty”.

Lines 435 & 436: It appears the 29 should be 30.

Lines 449-450: It would be useful to define what chemical classes FOP, DIM, and DEN represent.

Lines 454-455: Sentence is not complete.

Line 481: How is the 50-70% estimate derived?

Discussion

Line 501: How were the glaucophyte alga represented?

Line 503: Should the 71 be 710?

Lines 508-509: Statement about gene duplications needs a reference.

Line 522: Supplemental tables indicate crop plants also are resistant due to CYP clans 71 and 72 and GST classes Tau and Phi.

Lines 536-541: Are references needed for the statements concerning the different eras mentioned?

Figures/Tables

Figures 1-2: The colors are not clear/distinct; could this be improved?

Figure 2, line 262: Legend states 4 CYP clans are shown, but the figure shows 5. Maybe state increases in 4 of the 5 shown.

**Reviewer #2:** The manuscript describes the phylogenetic analysis of two types of enzymes that have key roles in non-target resistance to weeds from an evolutionary standpoint. The phylogenetic analysis appears to be very rigorous, but a slight weakness of the paper is around the descriptions of the basis of NTSR which are unclear in places.

Abstract

Line 32 ‘chemically alter’ is better described as ‘metabolism’ with these enzymes. This phrasing is repeated several times within the manuscript.

Introduction

Lines 46-52. The three sentences could do with redrafting as it comes across as unclear and repetitive.

Line 51 what is meant by ‘hyperactive forms of the enzymes’. I haven’t come across this terminology before.

Line 52 Ref. 5 is not a suitable reference in this context as it is an example of forced evolution in the laboratory. Are there any examples of enzymes with altered substrate specificity arising in nature?

Line 93 sentence should be amended to ‘CYPs and GSTs discovered so far that confer herbicide resistance.

Results

Table 1. Classification of Arabidopsis referred to as eudicot in table 1 but as a dicot in fig. 1. Would make comprehension easier if were consistent.

Fig.1 The key refers to the classification but in the text the actual species are discussed. This makes it difficult to understand unless you are very familiar with the classification categories. Either both or species should be added to the figure key.

Fig. 2 need to add units to the graph.

Line 322 Definition of ‘Archaeplastida’ would be useful for the more general reader.

Line 443 should refer to class for GSTs not clan

Discussion

Lines 524 -529. The enzymatic activity of these proteins is surely the most important reason as to why these clans and classes are important. Other CYPs and GSTs are up-regulated in NTSR populations but without activity towards herbicides are unlikely to have a major standalone effect on resistance.

Funding

Line 557 lacks grant number.

Supporting information

Table S3 Why are the 81As highlighted?

**Reviewer #3:** Genes encoding cytochrome P450 monooxygenases and glutathione S-transferases associated with herbicide resistance evolved before the origin of land plants

General Summary

The authors used publicly available genomic data to perform a phylogenetic analysis to uncover the origins of GST and CYPs in land plants. The discovered that certain clans within these enzymes were especially abundant in land plants while others were completely lost indicating that the expansion of these metabolic enzyme clans has a strong advantage for land plants. Furthermore, these expansion may help explain land plant evolution of non-target site resistance

General Comments

The writing was excellent. Very clear and thorough analysis. My biggest criticisms include

1) the results being redundant with much of the methods. Results can be tighter and reduced for readability.

2) Needs a better discussion of the Native functions of CYPs, from before herbicides were available. Land plant evolved much before agriculture so what drove certain clan expansion while others atrophied.

3) I think that perhaps some of the supplementary figures/tables, are quite important to understanding the discussion more completely. For instance, For me, Table S3 was valuable.

Line Comments

Ln 33: Please indicate briefly here that Cyp81s fall under the cyp 71 clan. Due to the importance of CYP81s in NTSR, this will be critical information to have upfront in the abstract for people who don’t read any further.

Ln 45-49: These sentences are largely redundant and can be condensed into a single statement.

Ln 49: I would imagine that the SNPs are not necessarily in the genes themselves, especially if they are changing the expression of a NTSR gene. Mutations in the gene body might modify NTSR protein ligand affinity, however, It would be more accurate to say genetic changes in the promoter of NTSR genes and/or the promoters of transcription factors that then in turn regulate NTSR gene expression are responsible for changing gene expression.

Ln 56: ‘Has’ should be ‘Have’

Ln 97-101: Missing in this hypothesis is the initial reason land plants would have more of these specific CYPs/GSTs. Surely clan number increase occurred well before the advent of herbicides. What evolutionary benefit would having more CYP/GST diversity in the clans provide in the absence of herbicides and why is it different for vascular plants then Archaeplastida.

Ln 155-156: Can there a description somewhere of what those ‘important residues’ might be? As is I, with my limited knowledge of cyp protein sequence, could not replicate your alignments and therefor your results.

Ln 213-223: There is a lot of re-hashing of methods in the results. It makes the results bloated. I suggest being more brief.

Ln 247-250: Is it possible that these 6 clans (71, 72, 74, 85, 86, and 727) evolved before the split of chlorophytes and streptophytes but were lost due to lack of selection? The divergence happen so long ago I could imagine lots of gene loss. I don’t know the phylogeny of land plants well enough to defend this hypothesis, I am curious if you have a reason to not favor this hypothesis.

Ln448-464: When mentioning the chemical classes it may be valuable for some readers to indicate what enzyme these chemistries inhibit or classify them by their HRAC code (Herbicide resistance Action Committee). Many weed scientists think about chemistries grouped like this.

Ln494-511: I would appreciate a discussion of native Cyp and GST function. The diversification of these clades in land plants was not driven by herbicides as it happened well before human agriculture. Maybe these clans are particularly useful in pathogen defense, antiherbivory, or chemical defense. Also, are plants with more CYPs in these clans more likely to evolve resistance? Could that be hypothesized and tested using similar phylogenetic approaches?

6. PLOS authors have the option to publish the peer review history of their article (what does this mean?). If published, this will include your full peer review and any attached files.

Reviewer #1: No

Reviewer #2: No

Reviewer #3: **Yes: **Eric Patterson

---

## [Author Response · Author response to Decision Letter 0]

28 Oct 2022

We are extremely grateful for the detailed, constructive and insightful comments of each of the three referees. We are also grateful for the errors that they identified in the previous version. Their recommendations have allowed us to improve the quality of our manuscript. Thank you.

Reviewer #1: General Comments:

This manuscript is well-written, clear, and organized. The topic offers a new perspective on herbicide resistance making the subject of interest to weed scientists as well as those interested in molecular phylogenetics. The article is a unique blend a literature review of NTS-based herbicide resistance and the role specific CYPs and GSTs play in that resistance.

Specific Comments/Questions:

Throughout the manuscript there is mention of hypotheses and how the data support various hypotheses (lines 266, 271, 368, 378, 390, 486, 489, 507). This hypothesis-testing emphasis allows the reader to read the document with the purpose in mind. However, it is unclear as to whether the hypotheses mentioned are the authors’ or the disciplines. In my opinion, I would state the hypotheses being addressed and tested up front in the introduction so that the reader can read the article with a goal in mind.

Response: 

We understand the ambiguity in our text identified by the referee. We have rewritten the highlighted sections to clarify that these are the authors’ hypotheses.

One improvement to the document would be to add more information on the phylogeny of the species selected for the analysis. Table 1 partially serves this purpose. Maybe Table 1 could be modified to include how these vascular and non-vascular plants and alga fit into the kingdom. Also, terms like embryophyte, bryophyte, glaucophyte, and Viridiplante are used without context or definition. Adding this information would make the manuscript an easier read for agronomists and weed scientists.

Response: 

We have made the changes recommended by the referee in Table 1 and in the text.

Another piece of information that would be helpful is the rationale used to determine which species to include in this analysis. First, why were these classes selected? For example, are ferns and conifers included? Also, of the 9 species selected for the analysis, why these? Is it due to genomic information or genome size? Are they recognized representatives of their classes in the field of plant phylogeny?

Response: 

We included rationales in the new version.

The authors imply that TSR and NTSR of weeds is due to mutations (first paragraph of introduction; Lines 462-463; Line 524). This is not always true. For example, could an herbicide molecule and a native substrate for a CYP or GST both fit in the enzyme’s active site? Also, some NTSR cases of weed resistance may be due to the accumulation of alleles that, when expressed, together confer tolerance to an herbicide (i.e., multigenic). If the term mutation is being more broadly used to include gene duplication or changes in gene regulation, it should be made clear.

Response:

The reviewer makes very good points. We have modified the text as suggested.

Finally, from the literature search and data analyses conducted, are the authors able to determine what percentage of NTSR cases are due to CYPs and/or GSTs? What percentage of metabolism-based resistance cases are not linked to any genes (unaccounted for)? Are there any linkages with the chemical classes subject to metabolism by GSTs and CYPs? In other words, are there any chemical motifs susceptible to metabolism by CYPs and GSTs? Determining the native function of these enzymes might enable this linkage.

Response:

We are unaware of any chemical motifs that are more susceptible to metabolism by GSTs or CYPs, as suggested by the reviewer. We mention this in the Discussion, to make this clear to the reader.

Minor Suggestions/Corrections:

Introduction

Line 44: Does “inhibit the interaction between the two” mean “reduce affinity of the herbicide for the target site”?

Response: 

Yes it does, we modified the revised text to reflect this. 

Line 72: Is there a description for Ure2p?

Response: 

Yes, we have included it in the revised text.

Lines 70-74: Is there a reference for this statement/lists?

Response: 

Yes, we have included them in this section.

Line 93: A brief definition of Archaeplastida would be helpful here.

Response: 

This has been included in the next paragraph of the revised text. 

Line 96: It would be helpful to define MYA for its first use.

Response: 

This has been added to the revised text.

Line 99: Since this is the first mention of K. nitens in the document, the genus name should be used.

Response: 

We included this change in the revised text. 

Materials & Methods

Line 116: Since this is the first mention of A. thaliana in the document, the genus name should be used. Also need a period at end of sentence.

Response: 

This has been corrected in the revised text.

Line 117: Was the ssp. japonica sequence used for Oryza sativa? If so, please indicate here. Figure 1 legend indicates the japonica sequence was used.

Response: 

Yes, this information was added to the revised text.

Line 118: Since this is the first mention of M. polymorpha in the document, the genus name should be used.

Response: 

This has been corrected in the revised version. 

Lines 116-127: Why is the protein sequence reference for S. moellendorffii not included with the other? 

Response: 

This information has been added to the revised text.

Line 146: Maybe change “don’t” to “do not”.

Response: 

This has been changed in the revised text. 

Results

Line 255: Define Viridiplantae here if not added earlier in the manuscript.

Response:

This has been added to the revised text and to Table 1. 

Line 280: Phrase “of the” repeated in sentence.

Response: 

This has been removed from the revised.

Line 283: For consistency use MYA?

Response: 

This has been modified in the revised text.

Line 284: Reference fig 2C?

Response: 

We have checked and corrected all references to fig 2C.

Line 284: Should this be four CYPs instead of 5 CYPs?

Response: 

Yes, this has been corrected in the revised text.

Line 303: Start sentence with “Sixteen”.

Response: 

This has been corrected in the revised text.

Line 323: Iota is shown in the Rhodophytes

Response: 

Yes, this manuscript identifies an Iota GST in C. merolae, indicating that Iota GSTs predate the divergence of rhodophytes and chlorophytes. 

Line 347: Start sentence with “Twelve”.

Response: 

This has been corrected in the revised text.

Lines 348-350: Is there a reference for this statement? There is no information in the manuscript to differentiate these 4 classes from the other 7 that also are shown as the earliest.

Response: 

I have included the missing information and references in this section of the revised text. 

Lines 399-401: How can these data account for there being only 1 Phi GST in S. moellendorffii? Please address.

Response: 

This has now been addressed in the paragraph of the revised text. 

Lines 427 & 448: “30” instead of “thirty”.

Response: 

This has been changed in the revised text.

Lines 435 & 436: It appears the 29 should be 30.

Response: 

That is correct, this was corrected in the revised text.

Lines 449-450: It would be useful to define what chemical classes FOP, DIM, and DEN represent.

Response: 

This information has been included in the revised text.

Lines 454-455: Sentence is not complete.

Response: 

This is corrected in the revised text.

Line 481: How is the 50-70% estimate derived?

Response: 

This estimate was derived from previously published GST phylogenies, however it has been amended to refer to the species in this paper. 

Discussion

Line 501: How were the glaucophyte alga represented?

Response: 

One Glaucophyte species nuclear genome has been sequenced to date, (Cyanophora paradoxa). It was an oversight not to include it.

Line 503: Should the 71 be 710?

Response: 

Yes, that is correct, and has been corrected in the revised text. 

Lines 508-509: Statement about gene duplications needs a reference.

Response: 

References 1 and 2 have been cited. 

Pégeot H, Koh CS, Petre B, Mathiot S, Duplessis S, Hecker A, et al. The poplar Phi class glutathione transferase: expression, activity and structure of GSTF1. Front Plant Sci. 2014;5:712. 

Werck-Reichhart D, Feyereisen R. Cytochromes P450: a success story. Genome Biol. 2000;1(6):1–9.

Line 522: Supplemental tables indicate crop plants also are resistant due to CYP clans 71 and 72 and GST classes Tau and Phi.

Response: 

Yes, this has been added to the revised text.

Lines 536-541: Are references needed for the statements concerning the different eras mentioned?

Response: 

Reference 131 has been inserted.

Cohen KM, Finney SC, Gibbard PL, Fan J-X. ICS International Chronostratigraphic Chart 2022/10 [Internet]. International Commission on Stratigraphy, IUGS. 2022 [cited 2022 Oct 16]. Available from: www.stratigraphy.org

Figures/Tables

Figures 1-2: The colors are not clear/distinct; could this be improved?

Response: 

Yes, the colours and figures have been made clearer in Figures 1 and 2. 

Figure 2, line 262: Legend states 4 CYP clans are shown, but the figure shows 5. Maybe state increases in 4 of the 5 shown.

Response: 

The figure legend has been changed accordingly in the revised text. 

Reviewer #2: The manuscript describes the phylogenetic analysis of two types of enzymes that have key roles in non-target resistance to weeds from an evolutionary standpoint. The phylogenetic analysis appears to be very rigorous, but a slight weakness of the paper is around the descriptions of the basis of NTSR which are unclear in places.

Abstract

Line 32 ‘chemically alter’ is better described as ‘metabolism’ with these enzymes. This phrasing is repeated several times within the manuscript.

Response: 

Chemically alter has been replaced with metabolise throughout the revised text.

Introduction

Lines 46-52. The three sentences could do with redrafting as it comes across as unclear and repetitive.

Response: 

They have been rewritten more concisely. 

Line 51 what is meant by ‘hyperactive forms of the enzymes’. I haven’t come across this terminology before.

Response: 

We have removed “hyperactive forms of the enzymes” from the text. This was included in error and we are grateful to the referee for spotting this.

Line 52 Ref. 5 is not a suitable reference in this context as it is an example of forced evolution in the laboratory. Are there any examples of enzymes with altered substrate specificity arising in nature?

Response: 

This reference has been removed. 

Line 93 sentence should be amended to ‘CYPs and GSTs discovered so far that confer herbicide resistance.

Response: 

This sentence has been amended as requested. 

Results

Table 1. Classification of Arabidopsis referred to as eudicot in table 1 but as a dicot in fig. 1. Would make comprehension easier if were consistent.

Response: 

This has been amended in the revised text.

Fig.1 The key refers to the classification but in the text the actual species are discussed. This makes it difficult to understand unless you are very familiar with the classification categories. Either both or species should be added to the figure key.

Response: 

The key has been modified to include species names in Figure 1. 

Fig. 2 need to add units to the graph.

Response: 

Units have been added to the revised Figure 2. 

Line 322 Definition of ‘Archaeplastida’ would be useful for the more general reader.

Response: 

A definition of Archaeplastida has been included in the Introduction and in Table 1. 

Line 443 should refer to class for GSTs not clan

Response: 

This has been corrected. 

Discussion

Lines 524 -529. The enzymatic activity of these proteins is surely the most important reason as to why these clans and classes are important. Other CYPs and GSTs are up-regulated in NTSR populations but without activity towards herbicides are unlikely to have a major standalone effect on resistance.

Response: 

Yes, this is a good point. However, it is unclear precisely what enzymatic activity unites the enzymes in these classes. This has now been included in the text between lines 537 and 555. 

Funding

Line 557 lacks a grant number.

Response: 

This has been amended. 

Supporting information

Table S3 Why are the 81As highlighted?

 Response: 

This was in error and the highlighting has been removed.

Reviewer #3: Genes encoding cytochrome P450 monooxygenases and glutathione S-transferases associated with herbicide resistance evolved before the origin of land plants

General Summary

The authors used publicly available genomic data to perform a phylogenetic analysis to uncover the origins of GST and CYPs in land plants. The discovered that certain clans within these enzymes were especially abundant in land plants while others were completely lost indicating that the expansion of these metabolic enzyme clans has a strong advantage for land plants. Furthermore, these expansion may help explain land plant evolution of non-target site resistance

General Comments

The writing was excellent. Very clear and thorough analysis. My biggest criticisms include

1) the results being redundant with much of the methods. Results can be tighter and reduced for readability.

Response: 

We have deleted redundant text. The Results of the revised manuscript are more concise than the original version. 

2) Needs a better discussion of the Native functions of CYPs, from before herbicides were available. Land plant evolved much before agriculture so what drove certain clan expansion while others atrophied.

Response: 

This is an important point and has been addressed in the Discussion (lines 535 to 555 and 563 to 570). 

3) I think that perhaps some of the supplementary figures/tables, are quite important to understanding the discussion more completely. For instance, For me, Table S3 was valuable.

Response: 

This a good point, and Table S3 and Table S4 have been included in the paper as Table 2 and Table 3. 

Line Comments

Ln 33: Please indicate briefly here that Cyp81s fall under the cyp 71 clan. Due to the importance of CYP81s in NTSR, this will be critical information to have upfront in the abstract for people who don’t read any further.

Response: 

This information has been included.

Ln 45-49: These sentences are largely redundant and can be condensed into a single statement.

Response: 

This section has been rewritten to be more concise.

Ln 49: I would imagine that the SNPs are not necessarily in the genes themselves, especially if they are changing the expression of a NTSR gene. Mutations in the gene body might modify NTSR protein ligand affinity, however, It would be more accurate to say genetic changes in the promoter of NTSR genes and/or the promoters of transcription factors that then in turn regulate NTSR gene expression are responsible for changing gene expression.

Response: 

This is a very good point and the text has been modified to reflect this.

Ln 56: ‘Has’ should be ‘Have’

Response: 

Overexpression is a singular noun and should be associated with a singular verb (has). “The overexpression of … has ….” therefore we have not changed “has” to “have” in this case.

Ln 97-101: Missing in this hypothesis is the initial reason land plants would have more of these specific CYPs/GSTs. Surely clan number increase occurred well before the advent of herbicides. What evolutionary benefit would having more CYP/GST diversity in the clans provide in the absence of herbicides and why is it different for vascular plants then Archaeplastida.

Response: 

This is an interesting point. We suspect that it is related to the increase in chemical and physiological diversity, but there is no evidence to support it. Therefore, we have not added a discussion of this point to the final sentence of the closing paragraph of the introduction. However, we added text that addresses this point at lines 563 to 565 of the Discussion.

“This expansion likely accompanied an increase in diversity of signalling molecules and secondary metabolites that may have occurred soon after plants started to grow in relatively dry terrestrial environments.”

Ln 155-156: Can there a description somewhere of what those ‘important residues’ might be? As is I, with my limited knowledge of cyp protein sequence, could not replicate your alignments and therefor your results.

Response: 

Yes, this is illustrated in supplementary figures S1, S3, and S4 however this wasn’t stated in the text. A sentence has now been included at this section referring to these figures.

Ln 213-223: There is a lot of re-hashing of methods in the results. It makes the results bloated. I suggest being more brief.

Response: 

This paragraph has been moved to Methods. 

Ln 247-250: Is it possible that these 6 clans (71, 72, 74, 85, 86, and 727) evolved before the split of chlorophytes and streptophytes but were lost due to lack of selection? The divergence happen so long ago I could imagine lots of gene loss. I don’t know the phylogeny of land plants well enough to defend this hypothesis, I am curious if you have a reason to not favor this hypothesis.

Response: 

It is possible but unlikely because these 6 clans were not present in the genomes of both C. reinhardtii and C. merolae. Therefore, the most parsimonious scenario is that these clans evolved in the last common ancestor of streptophytes. 

Ln448-464: When mentioning the chemical classes it may be valuable for some readers to indicate what enzyme these chemistries inhibit or classify them by their HRAC code (Herbicide resistance Action Committee). Many weed scientists think about chemistries grouped like this.

Response: 

This is a good point, HRAC codes have been included for each chemical class. 

Ln494-511: I would appreciate a discussion of native Cyp and GST function. The diversification of these clades in land plants was not driven by herbicides as it happened well before human agriculture. Maybe these clans are particularly useful in pathogen defense, antiherbivory, or chemical defense. Also, are plants with more CYPs in these clans more likely to evolve resistance? Could that be hypothesized and tested using similar phylogenetic approaches?

Response: 

This is a good point and we have inserted a paragraph between lines 535 and 555 that addresses this issue. However, there is little known and therefore our discussion is very speculative.

---

## [Decision Letter · Decision Letter 1]

10 Jan 2023

PONE-D-22-22418R1Genes encoding cytochrome P450 monooxygenases and glutathione *S*-transferases associated with herbicide resistance evolved before the origin of land plantsPLOS ONE

Dear Dr. Dolan,

Thank you for submitting your manuscript to PLOS ONE. After careful consideration, we believe your revised manuscript has scientific merit but does not fully meet PLOS ONE’s publication criteria as it currently stands. Therefore, we invite you to submit a revised version of the manuscript that addresses the following points raised during the review process.

            ==============================The revised manuscript is much improved; however, one previous reviewer and myself have provided additional comments and edits for you to consider to further improve the quality of the manuscript.Terminology regarding herbicide "tolerance" vs. "resistance" has not been used correctly; please see the accepted WSSA definitions and my comments below.Please consider alternative tolerance/resistance mechanisms other than CYP/GST expression. ============================== Please submit your revised manuscript by Feb 24 2023 11:59PM. If you will need more time than this to complete your revisions, please reply to this message or contact the journal office at plosone@plos.org. Please include the following items when submitting your revised manuscript:A rebuttal letter that responds to each point raised by the academic editor and reviewer. You should upload this letter as a separate file labeled 'Response to Reviewers'.A marked-up copy of your manuscript that highlights changes made to the original version. You should upload this as a separate file labeled 'Revised Manuscript with Track Changes'.An unmarked version of your revised paper without tracked changes. You should upload this as a separate file labeled 'Manuscript'.If applicable, we recommend that you deposit your laboratory protocols in protocols.io to enhance the reproducibility of your results. Protocols.io assigns your protocol its own identifier (DOI) so that it can be cited independently in the future. For instructions see: https://journals.plos.org/plosone/s/submission-guidelines#loc-laboratory-protocols. Additionally, PLOS ONE offers an option for publishing peer-reviewed Lab Protocol articles, which describe protocols hosted on protocols.io. Read more information on sharing protocols at https://plos.org/protocols?utm_medium=editorial-email&utm_source=authorletters&utm_campaign=protocols.

We look forward to receiving your revised manuscript.

Kind regards,

Dean E. Riechers, PhD

Academic Editor

PLOS ONE

Journal Requirements:

Additional Editor Comments:

**Academic Editor's synthesis comments.** Thank you for submitting your revised manuscript and detailed responses to the first three reviewers’ comments. One of the three original reviewers has again read and critiqued your revised manuscript and their comments are listed below. In addition, I have also read your original and revised manuscript thoroughly and have detailed below several additional comments and minor concerns that I’d like you to address in the new revised manuscript. If you can satisfactorily respond to and address these comments in your revision then I should be able to provide a quick turnaround. 

An important concern that should be fairly straightforward to address (without additional analysis) is that you are not using the terms herbicide tolerance and herbicide resistance correctly, per the Weed Science Society of America definitions of tolerance and resistance (see wssa.net for more information). Sometime you mention tolerance and resistance in the same sentence (e.g., lines 23-25 and 530-531) and sometimes you mention resistance when you actually should be stating tolerance. The WSSA definitions are as follows: (**a**) tolerance refers to a plant species that is not controlled at a herbicide rate that kills other species; tolerance is therefore considered “natural” and does not imply any type of selection pressure or artificial genetic manipulation, and (**b**) resistance refers to a plant or population of a species that is not controlled by a herbicide rate that typically kills plants within this same species; resistance therefore implies selection by a herbicide (or other factor) or artificial genetic manipulation, such as transgenic methods to make a GM crop variety. 

In many cases you wrote resistance when you are actually referring to a naturally tolerant crop or weed species. Examples of this are the majority of the crop species included in your CYP and GST analyses; these detoxification enzymes confer natural (or safener-induced) tolerance, and is therefore not involved with resistance and did not result from any known selection pressure. I do not believe this issue with your manuscript text is a deal breaker by any means, but I do believe you need to clearly rewrite portions of the manuscript and several table headings and figure captions. I realize that it would be much easier to include all CYPs and GSTs in your analysis under the “resistance” category, but unfortunately this is not scientifically accurate and could be misleading.

Other comments:

Line 147; reword “self-blasted” with a scientific term.

Lines 194-209; just an optional idea, but would it be informative to include the average and/or median number of GST and CYP genes in all populations you examined for comparison? If you disagree then that is fine, but please explain in your rebuttal comments.

Lines 384-86 and 407-409; I understand what you mean by writing about increases in CYPs and GSTs in certain species arising from specific clans or classes, but instead of “increase” would it be more accurate to say “enrichment” or another similar term? In other words, are you saying that the total number of protein-coding genes among species is about the same, but the number of CYPs and GSTs increased disproportionately? Maybe enrichment isn’t the best term either, but I think it might be worth explaining this concept in a bit more detail.

Lines 434-442, Tables 2 and 3, Tables S3 and S4; I have several comments here to consider and address: (**a**) as I mentioned earlier, it is not completely accurate to include all of these CYPs and GSTs under the category of resistance, since many of these genes and enzymes were discovered in naturally herbicide-tolerant crops – you could possibly change the title or heading to address this; (**b**) something that I believe would be interesting and important to discuss is the ratio of tau : phi GSTs in plant species where complete genomes are available. Several review articles have speculated about why the ratio of tau : phi GST varies among species (while total GSTs per genome also can vary slightly), but do you have any thoughts or insights on this varying ratio observation? For example, some reviews have speculated about different biochemical functions for tau vs phi plant GSTs; and (**c**) were your lists of GSTs and CYPs (regular in-text Tables and suppl. Tables for candidate genes) and references cited intended to be comprehensive via literature searches? I don’t normally like to do this as a journal peer reviewer or editor, but it seems unusual if you had comprehensively searched the literature that you would have missed several plant GST/CYP papers on tolerance/resistance published by my research group since the late 1990s. I will list these paper citations below for you to consider (either as verified detox gene/enzymes or candidates), and if you disagree with including them that is fine, but again please provide a justification in your rebuttal letter.

Baek, Y.S., Goodrich, L.V., et al. (2019). Front. Plant Sci. 10:192. doi:10.3389/fpls.2019.00192

Evans, A.F., O’Brien, S.R., et al. (2017). Plant Biotechnol. J. 15:1238-1249.

Pataky, J.K., Williams, M.M., et al. (2009). J. Amer. Soc. Hort. Sci. 134:252-260.

Nordby, J.N., Williams, M.M., et al. (2008). Weed Sci. 56:376-382.

Zhang, Q, Xu, F.-X., et al. (2007). Proteomics 7:1261-1278.

Zhang, Q., Riechers, D.E. (2004). Proteomics 4:2058-2071.

Xu, F.-X., Lagudah, E.S., et al. (2002). Plant Physiol. 130:362-373.

Riechers, D.E., Irzyk, G.P., et al. (1997). Plant Physiol. 114:1461-1470.

Line 476 and Figure 3; using white, light gray and dark gray bars is not ideal for differentiating the columns (bars) presented. If you do not want to include color, then can you at least use hatch marks or some other distinguishing feature besides shades of gray?

Line 478 and elsewhere in the manuscript; you briefly mention that CYP81 family members fall within the CYP71 clan, but I think you should clearly differentiate CYP clan vs family somewhere in the paper (perhaps even earlier in the introduction). The Hansen et al. 2021 CYP review has a nice box/figure that explains this topic, but I believe explaining this more in your paper will greatly help readers to understand the CYP nomenclature system…with can be confusing to expert and non-experts!

The two small paragraphs in lines 496-502 are poorly worded and confusing. For example, “mutate to herbicide resistance” is poorly worded and doesn’t make sense scientifically. Please rewrite these sentences and ideally merge into one cohesive paragraph.

Lines 512 and 516; please include HRAC group 15 at first mention of chloroacetanilides in line 512.

Lines 522-534; several issues with these sentences: (**1**) Do not start a paragraph/sentence with a number (522), (**2**) as mentioned earlier, I think the ratio of tau : phi class GSTs among species may be important to discuss and speculate about in addition to total number of GSTs per genome, (**3**) I would not consider A. thaliana to be a herbicide-tolerant weed…it is sensitive to many herbicides that kill dicots (530-31), and (**4**) the final brief paragraph basically restates what you’ve already written and should either be expanded, deleted, or merged with the previous paragraph.

Lines 539-540; I agree with the reviewer that you cannot always assume resistance or tolerance is due to a change in GST or CYP expression. It is possible that amino acid changes in the protein could enhance herbicide substrate affinity and/or enzyme catalytic detox efficiency with herbicide substrates. Please reword to include alternative explanations.

Lines 540-41; as I mentioned earlier, resistance indeed implies changes resulting from selection pressures, but you cannot include natural crop or weed tolerance in this category.

Lines 554-55; you cannot mix (i.e., use interchangeably) resistance and tolerance in the same sentence.

Lines 561-562; as noted by the reviewers, a lot of your Discussion section is redundant with the Results section. It seems as though I’m reading the same sentences numerous times throughout the manuscript. Please reword, condense, or delete as needed.

Line 566; I agree with the reviewer about possibly rewording “mutation” in light of gene copy number variation, gene duplication, etc.

Lines 583-587; one unifying feature of GST substrates (natural or xenobiotic) is an electrophilic carbon that can be attacked by the thiolate anion of GS-; please see our review article below (Riechers et al. 2010), but several other papers on GSTs have also mentioned this possible unifying factor.

Riechers, D.E., Kreuz, K. and Zhang, Q. (2010). Plant Physiol. 153:3-13.

Lines 606-08: this concluding sentence seems too strong and all-encompassing; please reword or soften a bit. As the reviewer also pointed out, I do not believe we can always assume that differences in CYP or GST gene expression account for natural tolerance or evolved weed resistance; for example, please consider coding region changes that may alter herbicide substrate specificity, enzyme activity, or other enzymatic detox functions that may also play a large role (or gene duplication, CNV, etc.). Even if examples have not yet been cited in the published literature, I think it is appropriate to speculate here.

Optional comment to consider about genes/enzymes other than CYPs or GSTs that govern crop tolerance: I believe one of the original reviewers mentioned or asked something about genes other than CYPs/GSTs involved in tolerance or resistance. It is purely up to you, but I thought I’d list that paper below that describes a novel type of oxygenase (HIS1; Fe(II)/2-oxoglutarate-dependent oxygenase) in rice that was discovered due to varietal sensitivity issues to HPPD inhibitors (Group 27 herbicides). Maeda, H., Murata, K., et al. (2019) A rice gene that confers broad-spectrum resistance to β-triketone herbicides. Science 365:393-396.

**Reviewers' comments: **

Reviewer's Responses to Questions

**Comments to the Author**

1. If the authors have adequately addressed your comments raised in a previous round of review and you feel that this manuscript is now acceptable for publication, you may indicate that here to bypass the “Comments to the Author” section, enter your conflict of interest statement in the “Confidential to Editor” section, and submit your "Accept" recommendation.

Reviewer #1: (No Response)

2. Is the manuscript technically sound, and do the data support the conclusions?

Reviewer #1: Yes

3. Has the statistical analysis been performed appropriately and rigorously? 

Reviewer #1: N/A

4. Have the authors made all data underlying the findings in their manuscript fully available?

Reviewer #1: Yes

5. Is the manuscript presented in an intelligible fashion and written in standard English?

Reviewer #1: Yes

6. Review Comments to the Author

**Reviewer #1: General Comments:**

The manuscript is an interesting analysis of the genes identified as responsible for metabolism-based NTS herbicide resistance in plants. The manuscript is well-written and should be of interest to a broad range of plant scientists.

Specific Comments/Questions:

The amendments to the tables, especially table 1, were useful in clarifying the relationship between the species selected for analysis. Major concerns of the reviewers appear to be addressed.

One area that is still unclear is where the diversity occurs among these genes identified as being responsible for herbicide metabolism (e.g., promoter region, substrate binding site). Supplementary figures 3 and 4 show sequence alignment, but could a summary statement as to how CYPs within a clan and GSTs within a class differ (i.e., what are the criteria used to distinguish one from another)? Are most of the differences in the substrate binding regions or in regulatory regions or both?

Minor Suggestions/Corrections:

Introduction

Lines 63 & 64: Capitalize and italicize the “s” in glutathione *S*-transferases.

Line 75: Space needed between “mitochondrial,or”.

Lines 102-107: The added sentences need a reference.

Materials & Methods

Lines 145-146: Why was the P. patens sequence used with A. thaliana and O. sativa for the GST BLASTP searches and not in the CYP BLASTP searches?

Line 156: Include “(2N)” after the words “N-terminal domains” to identify 2N.

Lines 153-158: The identifiers for the mitochondrial and microsomal GST classes do not match those in Lines 81-83 (Kappa vs metaxin class and MAPEG vs mPGES2 class).

Results

Lines 204-205: Why the distinction for the M. polymorpha CYP sequences here and no mention of other species?

Line 254: Maybe change “and” to “or” as all 6 species are not covered in each of the 4 (711, 727, 746, 747) of the 12 clans.

Lines 285 & 286: Should clan 741 be included in these two lists?

Line 324: For consistency, make streptophytes lower case.

Line 461: Reference Table 2 instead of S3.

Line 521: For completeness, add HRAC group for triazines.

Line 522: From which table or reference is the 61-80% derived?

Discussion

Line 566: Is “mutation” the correct word here? Isn’t it possible that these enzymes naturally metabolize the herbicide and that duplication of genes or copy numbers upon selection pressure confers the resistance?

Figures/Tables

Figure 1: It would be useful to add in the legend that chlorophyte is synonymous with streptophyte. Figures 1 & 2 use chlorophyte while the description in the results uses the streptophyte term.

Figure 2C: Should “phi” be included in the most wide-ranging list (all Archaeplastida) or with lambda, tau, and ure2p (land plants and charophytes/streptophytes)? The text in lines 359-361 indicates that “phi” should not be in the all-Archaeplastida list.

Table 3: Spelling of sulfonylurea varies between Tables 2 & 3.

7. PLOS authors have the option to publish the peer review history of their article (what does this mean?). If published, this will include your full peer review and any attached files.

Reviewer #1: No

---

## [Author Response · Author response to Decision Letter 1]

25 Jan 2023

Response to reviewers and editor comments:

We are extremely grateful for the detailed comments and concerns noted by the reviewers as they have further improved the quality of the manuscript and our understanding of the field. Please find below our responses to each of the comments in turn.

Additional Editor Comments:

Academic Editor's synthesis comments. Thank you for submitting your revised manuscript and detailed responses to the first three reviewers’ comments. One of the three original reviewers has again read and critiqued your revised manuscript and their comments are listed below. In addition, I have also read your original and revised manuscript thoroughly and have detailed below several additional comments and minor concerns that I’d like you to address in the new revised manuscript. If you can satisfactorily respond to and address these comments in your revision then I should be able to provide a quick turnaround. 

An important concern that should be fairly straightforward to address (without additional analysis) is that you are not using the terms herbicide tolerance and herbicide resistance correctly, per the Weed Science Society of America definitions of tolerance and resistance (see wssa.net for more information). Sometime you mention tolerance and resistance in the same sentence (e.g., lines 23-25 and 530-531) and sometimes you mention resistance when you actually should be stating tolerance. The WSSA definitions are as follows: (a) tolerance refers to a plant species that is not controlled at a herbicide rate that kills other species; tolerance is therefore considered “natural” and does not imply any type of selection pressure or artificial genetic manipulation, and (b) resistance refers to a plant or population of a species that is not controlled by a herbicide rate that typically kills plants within this same species; resistance therefore implies selection by a herbicide (or other factor) or artificial genetic manipulation, such as transgenic methods to make a GM crop variety. 

Response: we are grateful to the editor for pointing this out to us, and agree these terms should be used correctly and consistently. The recommended changes have been made to the manuscript. 

In many cases you wrote resistance when you are actually referring to a naturally tolerant crop or weed species. Examples of this are the majority of the crop species included in your CYP and GST analyses; these detoxification enzymes confer natural (or safener-induced) tolerance, and is therefore not involved with resistance and did not result from any known selection pressure. I do not believe this issue with your manuscript text is a deal breaker by any means, but I do believe you need to clearly rewrite portions of the manuscript and several table headings and figure captions. I realize that it would be much easier to include all CYPs and GSTs in your analysis under the “resistance” category, but unfortunately this is not scientifically accurate and could be misleading.

Response: we thank the editor for this comment and completely agree. The recommended changes have been made to the text, table headings and figure captions. 

Other comments:

Line 147; reword “self-blasted” with a scientific term.

Response: the sentence has been reworded. 

Lines 194-209; just an optional idea, but would it be informative to include the average and/or median number of GST and CYP genes in all populations you examined for comparison? If you disagree then that is fine, but please explain in your rebuttal comments.

Response: The average number of CYP genes identified in the 9 species is 126 sequences per species. The average number of GST genes identified in the 9 species is 40 sequences per species. This has been included in the paragraph. 

Lines 384-86 and 407-409; I understand what you mean by writing about increases in CYPs and GSTs in certain species arising from specific clans or classes, but instead of “increase” would it be more accurate to say “enrichment” or another similar term? In other words, are you saying that the total number of protein-coding genes among species is about the same, but the number of CYPs and GSTs increased disproportionately? Maybe enrichment isn’t the best term either, but I think it might be worth explaining this concept in a bit more detail.

Response: That is indeed what we mean. The sections of text have been reworded and expanded to hopefully explain this more clearly. 

Lines 434-442, Tables 2 and 3, Tables S3 and S4; I have several comments here to consider and address: (a) as I mentioned earlier, it is not completely accurate to include all of these CYPs and GSTs under the category of resistance, since many of these genes and enzymes were discovered in naturally herbicide-tolerant crops – you could possibly change the title or heading to address this; 

Response: this has been changed in the text and headings. 

(b) something that I believe would be interesting and important to discuss is the ratio of tau : phi GSTs in plant species where complete genomes are available. Several review articles have speculated about why the ratio of tau : phi GST varies among species (while total GSTs per genome also can vary slightly), but do you have any thoughts or insights on this varying ratio observation? For example, some reviews have speculated about different biochemical functions for tau vs phi plant GSTs; 

Response: this is a very interesting point. The ratio of Tau to Phi GSTs varies a lot between species and may be related to different biochemical functions fulfilling different environmental and ecological needs. This observation has been included in the results section in a new paragraph in lines 418 to 426 and in the discussion section in lines 547 to 549. 

and (c) were your lists of GSTs and CYPs (regular in-text Tables and suppl. Tables for candidate genes) and references cited intended to be comprehensive via literature searches? I don’t normally like to do this as a journal peer reviewer or editor, but it seems unusual if you had comprehensively searched the literature that you would have missed several plant GST/CYP papers on tolerance/resistance published by my research group since the late 1990s. I will list these paper citations below for you to consider (either as verified detox gene/enzymes or candidates), and if you disagree with including them that is fine, but again please provide a justification in your rebuttal letter.

Baek, Y.S., Goodrich, L.V., et al. (2019). Front. Plant Sci. 10:192. doi:10.3389/fpls.2019.00192

Evans, A.F., O’Brien, S.R., et al. (2017). Plant Biotechnol. J. 15:1238-1249.

Pataky, J.K., Williams, M.M., et al. (2009). J. Amer. Soc. Hort. Sci. 134:252-260.

Nordby, J.N., Williams, M.M., et al. (2008). Weed Sci. 56:376-382.

Zhang, Q, Xu, F.-X., et al. (2007). Proteomics 7:1261-1278.

Zhang, Q., Riechers, D.E. (2004). Proteomics 4:2058-2071.

Xu, F.-X., Lagudah, E.S., et al. (2002). Plant Physiol. 130:362-373.

Riechers, D.E., Irzyk, G.P., et al. (1997). Plant Physiol. 114:1461-1470.

Response: thank you for pointing out this oversight. They should be in the tables and have been included.

Line 476 and Figure 3; using white, light gray and dark gray bars is not ideal for differentiating the columns (bars) presented. If you do not want to include color, then can you at least use hatch marks or some other distinguishing feature besides shades of gray?

Response: the colours of the bars in Figure 3 have been improved. 

Line 478 and elsewhere in the manuscript; you briefly mention that CYP81 family members fall within the CYP71 clan, but I think you should clearly differentiate CYP clan vs family somewhere in the paper (perhaps even earlier in the introduction). The Hansen et al. 2021 CYP review has a nice box/figure that explains this topic, but I believe explaining this more in your paper will greatly help readers to understand the CYP nomenclature system…with can be confusing to expert and non-experts!

Response: this is a fair point and an explanation of CYP classification into clans and families has been included in the introduction. 

The two small paragraphs in lines 496-502 are poorly worded and confusing. For example, “mutate to herbicide resistance” is poorly worded and doesn’t make sense scientifically. Please rewrite these sentences and ideally merge into one cohesive paragraph.

Response: these paragraphs have been rewritten and merged. 

Lines 512 and 516; please include HRAC group 15 at first mention of chloroacetanilides in line 512.

Response: the HRAC group has been added. 

Lines 522-534; several issues with these sentences: (1) Do not start a paragraph/sentence with a number (522), (2) as mentioned earlier, I think the ratio of tau : phi class GSTs among species may be important to discuss and speculate about in addition to total number of GSTs per genome, (3) I would not consider A. thaliana to be a herbicide-tolerant weed…it is sensitive to many herbicides that kill dicots (530-31), and (4) the final brief paragraph basically restates what you’ve already written and should either be expanded, deleted, or merged with the previous paragraph.

Response: All recommended changes have been made to this paragraph. 

Lines 539-540; I agree with the reviewer that you cannot always assume resistance or tolerance is due to a change in GST or CYP expression. It is possible that amino acid changes in the protein could enhance herbicide substrate affinity and/or enzyme catalytic detox efficiency with herbicide substrates. Please reword to include alternative explanations.

Response: the sentence has been reworded. 

Lines 540-41; as I mentioned earlier, resistance indeed implies changes resulting from selection pressures, but you cannot include natural crop or weed tolerance in this category.

Response: this sentence has been reworded.

Lines 554-55; you cannot mix (i.e., use interchangeably) resistance and tolerance in the same sentence.

Response: Tolerance has been replaced with resistance in this sentence. 

Lines 561-562; as noted by the reviewers, a lot of your Discussion section is redundant with the Results section. It seems as though I’m reading the same sentences numerous times throughout the manuscript. Please reword, condense, or delete as needed.

Response: several sentences have been deleted.

Line 566; I agree with the reviewer about possibly rewording “mutation” in light of gene copy number variation, gene duplication, etc.

Response: this sentence has been reworded.

Lines 583-587; one unifying feature of GST substrates (natural or xenobiotic) is an electrophilic carbon that can be attacked by the thiolate anion of GS-; please see our review article below (Riechers et al. 2010), but several other papers on GSTs have also mentioned this possible unifying factor.

Riechers, D.E., Kreuz, K. and Zhang, Q. (2010). Plant Physiol. 153:3-13.

Response: This is a good point and has been included in the paragraph. 

Lines 606-08: this concluding sentence seems too strong and all-encompassing; please reword or soften a bit. As the reviewer also pointed out, I do not believe we can always assume that differences in CYP or GST gene expression account for natural tolerance or evolved weed resistance; for example, please consider coding region changes that may alter herbicide substrate specificity, enzyme activity, or other enzymatic detox functions that may also play a large role (or gene duplication, Copy Number Variation, etc.). Even if examples have not yet been cited in the published literature, I think it is appropriate to speculate here.

Response: the sentence has been softened and the other possible changes that could alter herbicide resistance/tolerance have been included. 

Optional comment to consider about genes/enzymes other than CYPs or GSTs that govern crop tolerance: I believe one of the original reviewers mentioned or asked something about genes other than CYPs/GSTs involved in tolerance or resistance. It is purely up to you, but I thought I’d list that paper below that describes a novel type of oxygenase (HIS1; Fe(II)/2-oxoglutarate-dependent oxygenase) in rice that was discovered due to varietal sensitivity issues to HPPD inhibitors (Group 27 herbicides). Maeda, H., Murata, K., et al. (2019) A rice gene that confers broad-spectrum resistance to β-triketone herbicides. Science 365:393-396.

Reviewers' comments: 

Reviewer's Responses to Questions

Comments to the Author

1. If the authors have adequately addressed your comments raised in a previous round of review and you feel that this manuscript is now acceptable for publication, you may indicate that here to bypass the “Comments to the Author” section, enter your conflict of interest statement in the “Confidential to Editor” section, and submit your "Accept" recommendation.

Reviewer #1: (No Response)

2. Is the manuscript technically sound, and do the data support the conclusions?

Reviewer #1: Yes

3. Has the statistical analysis been performed appropriately and rigorously?

Reviewer #1: N/A

4. Have the authors made all data underlying the findings in their manuscript fully available?

Reviewer #1: Yes

5. Is the manuscript presented in an intelligible fashion and written in standard English?

Reviewer #1: Yes

6. Review Comments to the Author

Reviewer #1: General Comments:

The manuscript is an interesting analysis of the genes identified as responsible for metabolism-based NTS herbicide resistance in plants. The manuscript is well-written and should be of interest to a broad range of plant scientists.

Specific Comments/Questions:

The amendments to the tables, especially table 1, were useful in clarifying the relationship between the species selected for analysis. Major concerns of the reviewers appear to be addressed.

One area that is still unclear is where the diversity occurs among these genes identified as being responsible for herbicide metabolism (e.g., promoter region, substrate binding site). Supplementary figures 3 and 4 show sequence alignment, but could a summary statement as to how CYPs within a clan and GSTs within a class differ (i.e., what are the criteria used to distinguish one from another)? Are most of the differences in the substrate binding regions or in regulatory regions or both?

Response:

Classification of CYPs into clans is based on their position on the tree. The variation in CYP protein sequences within a clan can be very high (less than 40% identity) throughout the alignment, except in the heme-binding, oxygen binding, and ERR triad domains which are more conserved. More variable regions include the membrane targeting region and substrate recognition sites. Sequence variation between clan members is higher in the larger clans. 

GSTs are classified into classes based on their sequence identity and kinetic properties. There is very large sequence diversity within classes in plants (can have lower than 40% amino acid identity within a class). The N-terminal domain containing the GSH-binding sites is usually more conserved and the C-terminal domain which contains most of the substrate recognition sites is more variable. 

Additional information on GST class and CYP clan/family classification has been provided in lines 69, and 91-94. An overview of protein features of GSTs has been added to S1 Figure. 

Minor Suggestions/Corrections:

Introduction

Lines 63 & 64: Capitalize and italicize the “s” in glutathione S-transferases.

Line 75: Space needed between “mitochondrial,or”.

Lines 102-107: The added sentences need a reference.

Response: The corrections have been made to the text. 

Materials & Methods

Lines 145-146: Why was the P. patens sequence used with A. thaliana and O. sativa for the GST BLASTP searches and not in the CYP BLASTP searches?

Response: A. thaliana and O. sativa CYP sequences were sufficient for the BLASTP searches to retrieve all CYP sequences in other species. 

In 2013 new plant GST classes were discovered in the genome of P. patens that do not exist in A. thaliana and O. sativa (hemerythrin, Iota and Ure2p) (Liu et al., 2013). The Hemerythrin and Iota GSTs have large class-specific protein domains and low amino acid sequence identity with sequences in other GST classes. For example, there is just 10% amino acid sequence identity between Pp3c3_21020V3.1 (PpGSTH1) and Pp3c15_23900V3.1 (PpGSTF1). To ensure that members of these classes were identified in the other species, P. patens GSTs were used as queries in addition to A. thaliana and O. sativa GSTs for BLASTP searches against the genomes of the other 8 species. 

Liu, Y. et al. Functional divergence of the glutathione S-transferase supergene family in Physcomitrella patens reveals complex patterns of large gene family evolution in land plants. (2013). Plant Physiology. 161(2): 773-86. doi: 10.1104/PP.112.205815.

Line 156: Include “(2N)” after the words “N-terminal domains” to identify 2N.

Response: the correction has been made to the text. 

Lines 153-158: The identifiers for the mitochondrial and microsomal GST classes do not match those in Lines 81-83 (Kappa vs metaxin class and MAPEG vs mPGES2 class).

Response: These are four different groups of proteins. Kappa and MAPEG proteins are sometimes called GSTs because they exhibit glutathione transferase activity, however they do not possess the characteristic GST domain architecture, and in the case of kappa GSTs, are more closely related to a bacterial protein disulphide bond isomerase (dsbA) than to other GST classes (Ladner et al., 2004). 

I have rewritten the sentence to include this information and referenced the two below papers that characterised kappa and MAPEG enzymes: 

Ladner, J., Parsons, J., et al. (2004). Parallel evolutionary pathways for glutathione transferases: structure and mechanism of the mitochondrial class kappa enzyme rGSTK1-1. Biochemistry. 43:352-61. doi: 10.1021/bi035832z.

Bresell, A., Weinander, R. et al. (2005). Bioinformatic and enzymatic characterization of the MAPEG superfamily. The FEBS Journal. 272:1688-1703. doi: 10.1111/J.1742-4658.2005.04596.X

Results

Lines 204-205: Why the distinction for the M. polymorpha CYP sequences here and no mention of other species?

Response: This is because they were named for this paper by Dr David Nelson who developed the naming system for cytochrome P450s. 

Line 254: Maybe change “and” to “or” as all 6 species are not covered in each of the 4 (711, 727, 746, 747) of the 12 clans.

Response: This has been corrected in the text. 

Lines 285 & 286: Should clan 741 be included in these two lists?

Response: Yes it should, it has been included. 

Line 324: For consistency, make streptophytes lower case.

Line 461: Reference Table 2 instead of S3.

Line 521: For completeness, add HRAC group for triazines.

Response: the suggested changes have been made to the text. 

Line 522: From which table or reference is the 61-80% derived?

Response: 61-80% is from the total number of GSTs identified in this study belonging to the tau and phi classes in S. moellendorffii, O. sativa and A. thaliana (61% in Sm, 67% in At, 80% in Os). The sentence has been rewritten for clarity. 

Discussion

Line 566: Is “mutation” the correct word here? Isn’t it possible that these enzymes naturally metabolize the herbicide and that duplication of genes or copy numbers upon selection pressure confers the resistance?

Response: this is true, the sentence has been reworded. 

Figures/Tables

Figure 1: It would be useful to add in the legend that chlorophyte is synonymous with streptophyte. Figures 1 & 2 use chlorophyte while the description in the results uses the streptophyte term.

Response: charophyte is synonymous with streptophyte algae. This has been included in the Fig 1 legend. 

Figure 2C: Should “phi” be included in the most wide-ranging list (all Archaeplastida) or with lambda, tau, and ure2p (land plants and charophytes/streptophytes)? The text in lines 359-361 indicates that “phi” should not be in the all-Archaeplastida list.

Response: this is correct and was written in error. Phi has been removed from the ‘all Archaeplastida’ list. 

Table 3: Spelling of sulfonylurea varies between Tables 2 & 3.

Response: thank you for noticing this, the spelling has been changed to be sulfonylurea. 

7. PLOS authors have the option to publish the peer review history of their article (what does this mean?). If published, this will include your full peer review and any attached files.

---

## [Editor Report · Decision Letter 2]

30 Jan 2023

PONE-D-22-22418R2Genes encoding cytochrome P450 monooxygenases and glutathione S-transferases associated with herbicide resistance evolved before the origin of land plantsPLOS ONE

Dear Dr. Dolan,

Thank you for submitting your manuscript to PLOS ONE. After careful consideration, the academic editor believes it has merit but does not fully meet PLOS ONE’s publication criteria as it currently stands. Therefore, we invite you to submit a revised version of the manuscript that addresses the minor points raised during the review process.

- new reference citations (#146-170) need to be included in the revised manuscript  

We look forward to receiving your revised manuscript.

Kind regards,

Dean E. Riechers, PhD

Academic Editor

PLOS ONE
---

## [Author Response · Author response to Decision Letter 2]

30 Jan 2023

Response to reviewers and editor:

We are extremely grateful for the detailed comments and concerns noted by the reviewers as they have further improved the quality of the manuscript and our understanding of the field. Please find below our responses to each of the comments in turn.

Additional Editor Comments:

Academic Editor's synthesis comments. Thank you for submitting your revised manuscript and detailed responses to the first three reviewers’ comments. One of the three original reviewers has again read and critiqued your revised manuscript and their comments are listed below. In addition, I have also read your original and revised manuscript thoroughly and have detailed below several additional comments and minor concerns that I’d like you to address in the new revised manuscript. If you can satisfactorily respond to and address these comments in your revision then I should be able to provide a quick turnaround. 

An important concern that should be fairly straightforward to address (without additional analysis) is that you are not using the terms herbicide tolerance and herbicide resistance correctly, per the Weed Science Society of America definitions of tolerance and resistance (see wssa.net for more information). Sometime you mention tolerance and resistance in the same sentence (e.g., lines 23-25 and 530-531) and sometimes you mention resistance when you actually should be stating tolerance. The WSSA definitions are as follows: (a) tolerance refers to a plant species that is not controlled at a herbicide rate that kills other species; tolerance is therefore considered “natural” and does not imply any type of selection pressure or artificial genetic manipulation, and (b) resistance refers to a plant or population of a species that is not controlled by a herbicide rate that typically kills plants within this same species; resistance therefore implies selection by a herbicide (or other factor) or artificial genetic manipulation, such as transgenic methods to make a GM crop variety. 

Response: we are grateful to the editor for pointing this out to us, and agree these terms should be used correctly and consistently. The recommended changes have been made to the manuscript. 

In many cases you wrote resistance when you are actually referring to a naturally tolerant crop or weed species. Examples of this are the majority of the crop species included in your CYP and GST analyses; these detoxification enzymes confer natural (or safener-induced) tolerance, and is therefore not involved with resistance and did not result from any known selection pressure. I do not believe this issue with your manuscript text is a deal breaker by any means, but I do believe you need to clearly rewrite portions of the manuscript and several table headings and figure captions. I realize that it would be much easier to include all CYPs and GSTs in your analysis under the “resistance” category, but unfortunately this is not scientifically accurate and could be misleading.

Response: we thank the editor for this comment and completely agree. The recommended changes have been made to the text, table headings and figure captions. 

Other comments:

Line 147; reword “self-blasted” with a scientific term.

Response: the sentence has been reworded. 

Lines 194-209; just an optional idea, but would it be informative to include the average and/or median number of GST and CYP genes in all populations you examined for comparison? If you disagree then that is fine, but please explain in your rebuttal comments.

Response: The average number of CYP genes identified in the 9 species is 126 sequences per species. The average number of GST genes identified in the 9 species is 40 sequences per species. This has been included in the paragraph. 

Lines 384-86 and 407-409; I understand what you mean by writing about increases in CYPs and GSTs in certain species arising from specific clans or classes, but instead of “increase” would it be more accurate to say “enrichment” or another similar term? In other words, are you saying that the total number of protein-coding genes among species is about the same, but the number of CYPs and GSTs increased disproportionately? Maybe enrichment isn’t the best term either, but I think it might be worth explaining this concept in a bit more detail.

Response: That is indeed what we mean. The sections of text have been reworded and expanded to hopefully explain this more clearly. 

Lines 434-442, Tables 2 and 3, Tables S3 and S4; I have several comments here to consider and address: (a) as I mentioned earlier, it is not completely accurate to include all of these CYPs and GSTs under the category of resistance, since many of these genes and enzymes were discovered in naturally herbicide-tolerant crops – you could possibly change the title or heading to address this; 

Response: this has been changed in the text and headings. 

(b) something that I believe would be interesting and important to discuss is the ratio of tau : phi GSTs in plant species where complete genomes are available. Several review articles have speculated about why the ratio of tau : phi GST varies among species (while total GSTs per genome also can vary slightly), but do you have any thoughts or insights on this varying ratio observation? For example, some reviews have speculated about different biochemical functions for tau vs phi plant GSTs; 

Response: this is a very interesting point. The ratio of Tau to Phi GSTs varies a lot between species and may be related to different biochemical functions fulfilling different environmental and ecological needs. This observation has been included in the results section in a new paragraph in lines 418 to 426 and in the discussion section in lines 547 to 549. 

and (c) were your lists of GSTs and CYPs (regular in-text Tables and suppl. Tables for candidate genes) and references cited intended to be comprehensive via literature searches? I don’t normally like to do this as a journal peer reviewer or editor, but it seems unusual if you had comprehensively searched the literature that you would have missed several plant GST/CYP papers on tolerance/resistance published by my research group since the late 1990s. I will list these paper citations below for you to consider (either as verified detox gene/enzymes or candidates), and if you disagree with including them that is fine, but again please provide a justification in your rebuttal letter.

Baek, Y.S., Goodrich, L.V., et al. (2019). Front. Plant Sci. 10:192. doi:10.3389/fpls.2019.00192

Evans, A.F., O’Brien, S.R., et al. (2017). Plant Biotechnol. J. 15:1238-1249.

Pataky, J.K., Williams, M.M., et al. (2009). J. Amer. Soc. Hort. Sci. 134:252-260.

Nordby, J.N., Williams, M.M., et al. (2008). Weed Sci. 56:376-382.

Zhang, Q, Xu, F.-X., et al. (2007). Proteomics 7:1261-1278.

Zhang, Q., Riechers, D.E. (2004). Proteomics 4:2058-2071.

Xu, F.-X., Lagudah, E.S., et al. (2002). Plant Physiol. 130:362-373.

Riechers, D.E., Irzyk, G.P., et al. (1997). Plant Physiol. 114:1461-1470.

Response: thank you for pointing out this oversight. They should be in the tables and have been included.

Line 476 and Figure 3; using white, light gray and dark gray bars is not ideal for differentiating the columns (bars) presented. If you do not want to include color, then can you at least use hatch marks or some other distinguishing feature besides shades of gray?

Response: the colours of the bars in Figure 3 have been improved. 

Line 478 and elsewhere in the manuscript; you briefly mention that CYP81 family members fall within the CYP71 clan, but I think you should clearly differentiate CYP clan vs family somewhere in the paper (perhaps even earlier in the introduction). The Hansen et al. 2021 CYP review has a nice box/figure that explains this topic, but I believe explaining this more in your paper will greatly help readers to understand the CYP nomenclature system…with can be confusing to expert and non-experts!

Response: this is a fair point and an explanation of CYP classification into clans and families has been included in the introduction. 

The two small paragraphs in lines 496-502 are poorly worded and confusing. For example, “mutate to herbicide resistance” is poorly worded and doesn’t make sense scientifically. Please rewrite these sentences and ideally merge into one cohesive paragraph.

Response: these paragraphs have been rewritten and merged. 

Lines 512 and 516; please include HRAC group 15 at first mention of chloroacetanilides in line 512.

Response: the HRAC group has been added. 

Lines 522-534; several issues with these sentences: (1) Do not start a paragraph/sentence with a number (522), (2) as mentioned earlier, I think the ratio of tau : phi class GSTs among species may be important to discuss and speculate about in addition to total number of GSTs per genome, (3) I would not consider A. thaliana to be a herbicide-tolerant weed…it is sensitive to many herbicides that kill dicots (530-31), and (4) the final brief paragraph basically restates what you’ve already written and should either be expanded, deleted, or merged with the previous paragraph.

Response: All recommended changes have been made to this paragraph. 

Lines 539-540; I agree with the reviewer that you cannot always assume resistance or tolerance is due to a change in GST or CYP expression. It is possible that amino acid changes in the protein could enhance herbicide substrate affinity and/or enzyme catalytic detox efficiency with herbicide substrates. Please reword to include alternative explanations.

Response: the sentence has been reworded. 

Lines 540-41; as I mentioned earlier, resistance indeed implies changes resulting from selection pressures, but you cannot include natural crop or weed tolerance in this category.

Response: this sentence has been reworded.

Lines 554-55; you cannot mix (i.e., use interchangeably) resistance and tolerance in the same sentence.

Response: Tolerance has been replaced with resistance in this sentence. 

Lines 561-562; as noted by the reviewers, a lot of your Discussion section is redundant with the Results section. It seems as though I’m reading the same sentences numerous times throughout the manuscript. Please reword, condense, or delete as needed.

Response: several sentences have been deleted.

Line 566; I agree with the reviewer about possibly rewording “mutation” in light of gene copy number variation, gene duplication, etc.

Response: this sentence has been reworded.

Lines 583-587; one unifying feature of GST substrates (natural or xenobiotic) is an electrophilic carbon that can be attacked by the thiolate anion of GS-; please see our review article below (Riechers et al. 2010), but several other papers on GSTs have also mentioned this possible unifying factor.

Riechers, D.E., Kreuz, K. and Zhang, Q. (2010). Plant Physiol. 153:3-13.

Response: This is a good point and has been included in the paragraph. 

Lines 606-08: this concluding sentence seems too strong and all-encompassing; please reword or soften a bit. As the reviewer also pointed out, I do not believe we can always assume that differences in CYP or GST gene expression account for natural tolerance or evolved weed resistance; for example, please consider coding region changes that may alter herbicide substrate specificity, enzyme activity, or other enzymatic detox functions that may also play a large role (or gene duplication, Copy Number Variation, etc.). Even if examples have not yet been cited in the published literature, I think it is appropriate to speculate here.

Response: the sentence has been softened and the other possible changes that could alter herbicide resistance/tolerance have been included. 

Optional comment to consider about genes/enzymes other than CYPs or GSTs that govern crop tolerance: I believe one of the original reviewers mentioned or asked something about genes other than CYPs/GSTs involved in tolerance or resistance. It is purely up to you, but I thought I’d list that paper below that describes a novel type of oxygenase (HIS1; Fe(II)/2-oxoglutarate-dependent oxygenase) in rice that was discovered due to varietal sensitivity issues to HPPD inhibitors (Group 27 herbicides). Maeda, H., Murata, K., et al. (2019) A rice gene that confers broad-spectrum resistance to β-triketone herbicides. Science 365:393-396.

Reviewers' comments: 

Reviewer's Responses to Questions

Comments to the Author

1. If the authors have adequately addressed your comments raised in a previous round of review and you feel that this manuscript is now acceptable for publication, you may indicate that here to bypass the “Comments to the Author” section, enter your conflict of interest statement in the “Confidential to Editor” section, and submit your "Accept" recommendation.

Reviewer #1: (No Response)

2. Is the manuscript technically sound, and do the data support the conclusions?

Reviewer #1: Yes

3. Has the statistical analysis been performed appropriately and rigorously?

Reviewer #1: N/A

4. Have the authors made all data underlying the findings in their manuscript fully available?

Reviewer #1: Yes

5. Is the manuscript presented in an intelligible fashion and written in standard English?

Reviewer #1: Yes

6. Review Comments to the Author

Reviewer #1: General Comments:

The manuscript is an interesting analysis of the genes identified as responsible for metabolism-based NTS herbicide resistance in plants. The manuscript is well-written and should be of interest to a broad range of plant scientists.

Specific Comments/Questions:

The amendments to the tables, especially table 1, were useful in clarifying the relationship between the species selected for analysis. Major concerns of the reviewers appear to be addressed.

One area that is still unclear is where the diversity occurs among these genes identified as being responsible for herbicide metabolism (e.g., promoter region, substrate binding site). Supplementary figures 3 and 4 show sequence alignment, but could a summary statement as to how CYPs within a clan and GSTs within a class differ (i.e., what are the criteria used to distinguish one from another)? Are most of the differences in the substrate binding regions or in regulatory regions or both?

Response:

Classification of CYPs into clans is based on their position on the tree. The variation in CYP protein sequences within a clan can be very high (less than 40% identity) throughout the alignment, except in the heme-binding, oxygen binding, and ERR triad domains which are more conserved. More variable regions include the membrane targeting region and substrate recognition sites. Sequence variation between clan members is higher in the larger clans. 

GSTs are classified into classes based on their sequence identity and kinetic properties. There is very large sequence diversity within classes in plants (can have lower than 40% amino acid identity within a class). The N-terminal domain containing the GSH-binding sites is usually more conserved and the C-terminal domain which contains most of the substrate recognition sites is more variable. 

Additional information on GST class and CYP clan/family classification has been provided in lines 69, and 91-94. An overview of protein features of GSTs has been added to S1 Figure. 

Minor Suggestions/Corrections:

Introduction

Lines 63 & 64: Capitalize and italicize the “s” in glutathione S-transferases.

Line 75: Space needed between “mitochondrial,or”.

Lines 102-107: The added sentences need a reference.

Response: The corrections have been made to the text. 

Materials & Methods

Lines 145-146: Why was the P. patens sequence used with A. thaliana and O. sativa for the GST BLASTP searches and not in the CYP BLASTP searches?

Response: A. thaliana and O. sativa CYP sequences were sufficient for the BLASTP searches to retrieve all CYP sequences in other species. 

In 2013 new plant GST classes were discovered in the genome of P. patens that do not exist in A. thaliana and O. sativa (hemerythrin, Iota and Ure2p) (Liu et al., 2013). The Hemerythrin and Iota GSTs have large class-specific protein domains and low amino acid sequence identity with sequences in other GST classes. For example, there is just 10% amino acid sequence identity between Pp3c3_21020V3.1 (PpGSTH1) and Pp3c15_23900V3.1 (PpGSTF1). To ensure that members of these classes were identified in the other species, P. patens GSTs were used as queries in addition to A. thaliana and O. sativa GSTs for BLASTP searches against the genomes of the other 8 species. 

Liu, Y. et al. Functional divergence of the glutathione S-transferase supergene family in Physcomitrella patens reveals complex patterns of large gene family evolution in land plants. (2013). Plant Physiology. 161(2): 773-86. doi: 10.1104/PP.112.205815.

Line 156: Include “(2N)” after the words “N-terminal domains” to identify 2N.

Response: the correction has been made to the text. 

Lines 153-158: The identifiers for the mitochondrial and microsomal GST classes do not match those in Lines 81-83 (Kappa vs metaxin class and MAPEG vs mPGES2 class).

Response: These are four different groups of proteins. Kappa and MAPEG proteins are sometimes called GSTs because they exhibit glutathione transferase activity, however they do not possess the characteristic GST domain architecture, and in the case of kappa GSTs, are more closely related to a bacterial protein disulphide bond isomerase (dsbA) than to other GST classes (Ladner et al., 2004). 

I have rewritten the sentence to include this information and referenced the two below papers that characterised kappa and MAPEG enzymes: 

Ladner, J., Parsons, J., et al. (2004). Parallel evolutionary pathways for glutathione transferases: structure and mechanism of the mitochondrial class kappa enzyme rGSTK1-1. Biochemistry. 43:352-61. doi: 10.1021/bi035832z.

Bresell, A., Weinander, R. et al. (2005). Bioinformatic and enzymatic characterization of the MAPEG superfamily. The FEBS Journal. 272:1688-1703. doi: 10.1111/J.1742-4658.2005.04596.X

Results

Lines 204-205: Why the distinction for the M. polymorpha CYP sequences here and no mention of other species?

Response: This is because they were named for this paper by Dr David Nelson who developed the naming system for cytochrome P450s. 

Line 254: Maybe change “and” to “or” as all 6 species are not covered in each of the 4 (711, 727, 746, 747) of the 12 clans.

Response: This has been corrected in the text. 

Lines 285 & 286: Should clan 741 be included in these two lists?

Response: Yes it should, it has been included. 

Line 324: For consistency, make streptophytes lower case.

Line 461: Reference Table 2 instead of S3.

Line 521: For completeness, add HRAC group for triazines.

Response: the suggested changes have been made to the text. 

Line 522: From which table or reference is the 61-80% derived?

Response: 61-80% is from the total number of GSTs identified in this study belonging to the tau and phi classes in S. moellendorffii, O. sativa and A. thaliana (61% in Sm, 67% in At, 80% in Os). The sentence has been rewritten for clarity. 

Discussion

Line 566: Is “mutation” the correct word here? Isn’t it possible that these enzymes naturally metabolize the herbicide and that duplication of genes or copy numbers upon selection pressure confers the resistance?

Response: this is true, the sentence has been reworded. 

Figures/Tables

Figure 1: It would be useful to add in the legend that chlorophyte is synonymous with streptophyte. Figures 1 & 2 use chlorophyte while the description in the results uses the streptophyte term.

Response: charophyte is synonymous with streptophyte algae. This has been included in the Fig 1 legend. 

Figure 2C: Should “phi” be included in the most wide-ranging list (all Archaeplastida) or with lambda, tau, and ure2p (land plants and charophytes/streptophytes)? The text in lines 359-361 indicates that “phi” should not be in the all-Archaeplastida list.

Response: this is correct and was written in error. Phi has been removed from the ‘all Archaeplastida’ list. 

Table 3: Spelling of sulfonylurea varies between Tables 2 & 3.

Response: thank you for noticing this, the spelling has been changed to be sulfonylurea. 

7. PLOS authors have the option to publish the peer review history of their article (what does this mean?). If published, this will include your full peer review and any attached files.

---

## [Editor Report · Decision Letter 3]

7 Feb 2023

Genes encoding cytochrome P450 monooxygenases and glutathione *S*-transferases associated with herbicide resistance evolved before the origin of land plants

PONE-D-22-22418R3

Dear Dr. Dolan,

We’re pleased to inform you that your manuscript has been judged scientifically suitable for publication and will be formally accepted for publication once it meets all outstanding technical requirements.

Kind regards,

Dean E. Riechers, PhD

Academic Editor

PLOS ONE
---

## [Editor Report · Acceptance letter]

9 Feb 2023

PONE-D-22-22418R3 

Genes encoding cytochrome P450 monooxygenases and glutathione S-transferases associated with herbicide resistance evolved before the origin of land plants 

Dear Dr. Dolan:

I'm pleased to inform you that your manuscript has been deemed suitable for publication in PLOS ONE. Congratulations! Your manuscript is now with our production department. 

Kind regards, 

on behalf of

Dr. Dean E. Riechers 

Academic Editor

PLOS ONE